# Neural representation of goal direction in the monarch butterfly brain

**M. Jerome Beetz** [1] ✉, **Christian Kraus** [1,2] **& Basil el Jundi** [1,2]

Neural processing of a desired moving direction requires the continuous comparison between the current heading and the goal direction. While the neural basis underlying the current heading is well-studied, the coding of the goal direction remains unclear in insects. Here, we used tetrode recordings in tethered flying monarch butterflies to unravel how a goal direction is represented in the insect brain. While recording, the butterflies maintained robust goal directions relative to a virtual sun. By resetting their goal directions, we found neurons whose spatial tuning was tightly linked to the goal directions. Importantly, their tuning was unaffected when the butterflies changed their heading after compass perturbations, showing that these neurons specifically encode the goal direction. Overall, we here discovered invertebrate goal-direction neurons that share functional similarities to goal-direction cells reported in mammals. Our results give insights into the evolutionarily conserved principles of goal-directed spatial orientation in animals.

To maintain a constant moving direction, animals register their current orientation in space, as well as the intended goal direction. Consequently, their brain constantly compares the current heading direction with an internally represented goal direction[1–3]. If these two directions mismatch, the brain generates steering commands that return the animal to its goal direction. While the heading direction is encoded by evolutionarily conserved head-direction (HD) neurons found in diverse species ranging from invertebrates[4–6] to vertebrates[7–14], goal-direction (GD) neurons have only been empirically reported in the mammalian brain[15–21]. The action potential rate of these GD neurons strongly correlates with the animal's bearing relative to the goal[15]. Although insect GD neurons have been predicted in several theoretical studies[22–27] it is still not fully clear how the goal direction is encoded in insects. A robust representation of the goal direction, however, is of highest ecological importance for migratory insects such as monarch butterflies that seasonally migrate up to 5.000 kilometers from southern Canada and the northern United States to their overwintering site in central Mexico[28,29]. By using the sun as their main orientation reference, the butterflies constantly keep track of their southward compass direction[30,31]. In the brain, sun compass information is processed in the central complex[32–35]. Given the highly conserved function of the insect central complex[1,36–38], this brain region is central for

spatial orientation in monarch butterflies. Recent neurophysiological experiments on tethered flying butterflies have shown that the butterfly central complex houses HD neurons that, consistent with findings in other insect species[39–41], infer the current heading from sun compass information[6]. In addition, findings in cockroaches and fruit flies have shown that the central complex houses pre-motor neurons that transmit commands to descending pathways for controlling the steering direction[42–44]. To be able to control for steering, the central complex needs additional information about the insect's goal direction as proposed in anatomical and computational studies[22,23,25–27,43,45,46]. Because manipulations of the neural activity in HD neurons did not affect the goal direction in fruit flies[3], goal directions must be processed by different central-complex neurons. Notably, a recent study shows that the activity of a set of central-complex neurons triggers a goal-directed orientation in fruit flies[43]. However, to what extent the activity of these neurons dictates the fly's goal direction remains unclear, so far[43,47].

To identify GD neurons in the insect brain, we performed long-term tetrode recordings in tethered flying monarch butterflies. During the recordings, the butterflies maintained a goal direction with respect to a virtual sun. This experimental design enabled us to monitor the neural activity of functionally different types of central-complex

[1]Zoology II, Biocenter, University of Würzburg, Würzburg, Germany. [2]Animal Physiology, Department of Biology, Norwegian University of Science and Technology, Trondheim, Norway. ✉e-mail: jerome.beetz@uni-wuerzburg.de

neurons while selectively manipulating the compass or goal direction of actively orienting butterflies. As expected, we found HD neurons that represented the butterflies' compass but not their goal direction. Remarkably, another subset of neurons was specifically tuned to the butterflies' goal-direction: When we conditioned the butterflies to set a new goal direction, the angular tuning of these neurons accurately followed the behavioral change in goal direction implying that these neurons represent GD neurons. Moreover, by simultaneously recording from GD and steering neurons, we found that the GD neurons are suited to initiate turns back to the goal heading whenever the animal was facing away from its goal direction. Despite being evolutionarily distant, our results show that the insect central complex houses GD neurons similar to the ones described in the mammalian brain[15], highlighting the computational power of the insect brain in goal-directed spatial orientation.

## Results

### Monarch butterflies maintain a goal direction relative to a virtual sun

We tethered monarch butterflies at the center of a flight simulator, in which they could freely steer in any goal direction with respect to a virtual sun (Fig. 1a). Although the tested monarch butterflies were non-migratory, they reliably maintained goal directions (Fig. 1b), consistent with recent behavioral findings[48]. Across individuals, the goal directions were arbitrary (Fig. S1) resembling menotactic orientation behavior observed in a variety of insects including dung beetles[49] and fruit flies[40,50]. Menotactic orientation optimizes animal dispersal[49] and can be expected to emerge by matching the current heading—encoded by the butterflies' compass—with an internal goal representation[2,3]. To dissociate between compass and goal direction coding, we selectively perturbed the butterflies' compass without affecting their intended goal direction (Fig. 1c). This was achieved by displacing the virtual sun along the azimuth at different angular sizes, i.e., 180°, 90°, 45°, 25°, and 15°, every 90 s (Fig. 1c, S2). To maintain the initial goal direction relative to the sun, individual butterflies adjusted their heading direction in accordance with the new sun position. A sun displacement of 180° evoked a 180° change in the butterflies' flight direction, resulting in a similar goal direction relative to the virtual sun after compass perturbation (Fig. 1d). To quantify how well the butterflies ($N = 32$) followed the sun displacements of different angular sizes, we calculated the difference in mean heading (magenta lines in Fig. 1d) prior to and after each sun displacement (Fig. S3a–e). On average, the butterflies' change in flight direction was highly associated with the displacements of the virtual sun (mean ± standard deviation: $0.91 \pm 0.4$; Fig. 1e). Consistently, the angular size of sun displacements subtracted from the change in heading direction was clustered around 0° (Fig. S3f). Taken together, these data suggest that the butterflies used the virtual sun to maintain a goal direction in our setup. Moreover, our data show that we successfully shifted the polarity of the butterflies' compass through sun displacements (compass perturbation), without affecting the goal direction. Thus, we expected that neurons representing compass information, such as HD neurons, should change their spatial tuning following compass perturbations, contrasted by GD neurons whose spatial tuning should remain invariant.

### Compass perturbations revealed putative GD neurons

While perturbing the butterflies' compass system, we monitored the neural activity of the central complex (Figs. 1f, S4). We recorded from 113 neurons ($-4.6 \pm 2.2$ neurons/animal) that showed a persistent angular tuning in darkness (Fig. S5a, b). This directional coding in the absence of visual information shows that these neurons maintain an internal representation of the directional information, as expected from HD and GD neurons[4,6,51]. The tuning directedness to the virtual sun, represented by the mean vector length (MVL; mean ± standard deviation: $0.15 \pm 0.09$; Fig. S5c), was statistically longer than the tuning

directedness modeled from shuffled data (Fig. 1g, $p < 10^{-5}$, W = −10878, $n = 113$, two-sided Wilcoxon matched-pairs signed-rank test), suggesting that the neurons exhibited a directed angular tuning during flight.

Based on the influence of compass perturbations on the neural tuning, we classified the neurons into two types: (i) compass neurons whose angular tuning was linked to the butterflies' heading direction, such as HD and steering neurons, and (ii) putative GD neurons whose angular tuning was not affected by compass perturbations. The classification was quantified by calculating an HD index for each neuron (for details, see "Methods", Fig. S6). Positive HD indices were expected from compass neurons that change their angular tuning, represented by the preferred firing direction (pfd), in accordance with the butterfly's change in mean heading (green neurons in Fig. 1h and Figs. S7a, S6a). In contrast, putative GD neurons should show negative HD indices as their angular tunings were expected to be unaffected by compass perturbations (blue neurons in Fig. 1h and Fig. S7a; Fig. S6b). In total, 55 of 113 neurons (48.7%) were classified as compass neurons (HD index: mean ± standard deviation: $0.38 \pm 0.29$, Fig. S6c). Their angular tuning changed after compass perturbations if visualized in an absolute frame of reference (0° represents a fixed direction in the setup; upper heatmaps in Fig. 1i). Neither variations in their action potential rate during flight nor their mean spike rate could explain the observed tuning changes ($p = 0.75$, U = 1540; two-sided Mann–Whitney U test, Fig. S8). The strong association between the animal's heading and spatial tuning of compass neurons is apparent when the neurons' firing rate is plotted relative to the butterflies' mean heading (0° represents the animal's heading direction; Fig. S7b). In contrast to this, 58 neurons (51.3%) had negative HD indices (mean ± standard deviation: $-0.43 \pm 0.32$) and might, amongst others, include neurons that represent the animal's goal direction (lower heatmaps in Fig. 1i). The correlation between their angular tuning measured before and after compass perturbations was much higher than that of compass neurons ($p = 0.001$, U = 1027, two-sided Mann–Whitney U test, Fig. 1j). Consistent with this, the pfds of putative GD neurons varied less than those of compass neurons ($p < 0.0001$, U = 494, two-sided Mann–Whitney U, Fig. S7c). Moreover, the tuning of these putative GD neurons showed a higher variance of heading offsets than the compass neurons ($p = 0.0054$, U = 1113, two-sided Mann–Whitney U test, Fig. 1k) indicating that their angular tuning was not linked to the coding of the butterflies' compass. Both compass neurons and putative GD neurons fully tiled a 360° representation of angular space (compass neurons: $p = 0.76$, Z = 0.27; $n = 55$; putative GD neurons: $p = 0.36$, Z = 1.01; $n = 58$ Rayleigh test, Fig. 1l). Altogether, the compass perturbations allowed us to functionally discriminate between two types of neurons, one type that was closely associated with the heading coding (compass neurons) and another type whose spatial tuning was invariant in response to compass perturbations (putative GD neurons).

### Resetting the butterflies' goal directions

At this point, it was unclear whether the putative GD neurons truly encoded the animals' goal direction or any stable cue in the environment, such as magnetic information[52]. To eliminate this uncertainty and ultimately test for goal-direction coding, we next reset the goal direction of 17 butterflies—following compass perturbations—by applying small electric shocks to the butterflies' necks whenever they headed towards their initial goal direction (±90°; Fig. 2a). This aversive conditioning indeed reliably changed the butterflies' goal direction ($129.7° \pm 39.9°$; $p = 0.007$, W = 9.832, Mardia-Watson-Wheeler test; Fig. 2b, c; compare Supplementary Movie 1 showing pre-conditioning with Supplementary Movie 2 showing post-conditioning). After conditioning, most butterflies (13 of 17) headed into the hemisphere opposite to the virtual sun (Fig. S9). This is not surprising considering that most of the 17 butterflies set their goal direction toward the virtual sun hemisphere prior to conditioning, a trend often observed in indoor experiments[40,53,54]. However, the butterflies' headings showed a

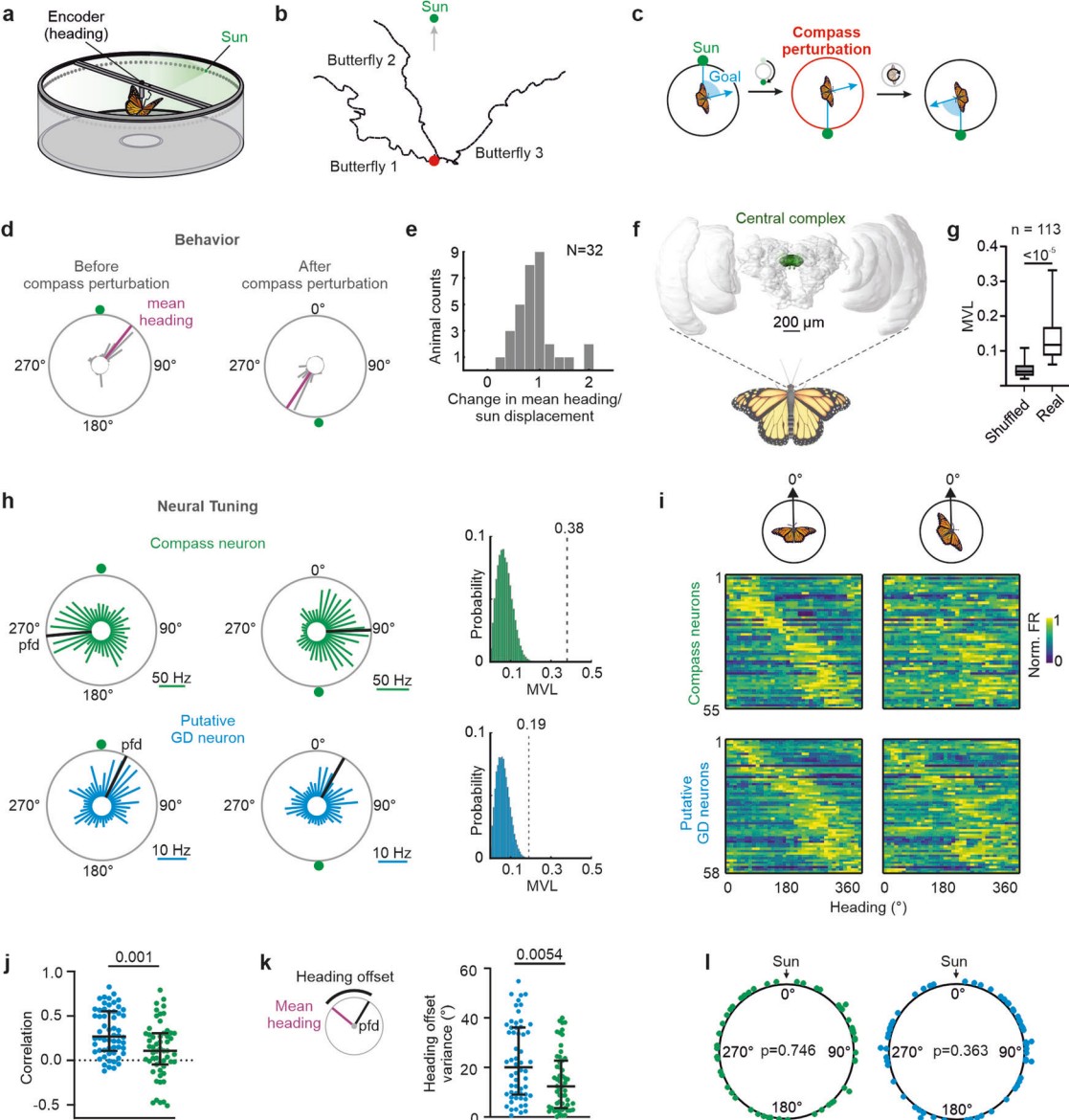

**Fig. 1 | Perturbation of the butterfly compass. a** Schematic drawing of the flight simulator (modified from ref. 6). **b** Virtual flight trajectories of three butterflies. The red dot represents the starting point. **c** For a consistent goal direction relative to the sun, butterflies follow a sun displacement of e.g., 180°. **d** The flight heading of one butterfly before and after sun displacement. Magenta lines represent the mean heading. **e** The butterflies ($N = 32$) followed the sun displacement as indicated by ratios between change in heading and sun displacement clustering around one. **f** The butterfly brain (anterior view[78];) with the central complex highlighted. **g** The neurons' mean vector length (MVL) of tuning was longer than for shuffled data ($n = 113$, $p < 10^{-5}$, W = −10878, two-sided Wilcoxon test). Boxes and whiskers show 25th/75th and 5th/95th percentile respectively and the median. **h** Angular tuning of two neurons (green and blue) before (left) and after (right) a 180° sun displacement. Black lines indicate the neurons' preferred firing directions (pfds). (Right) The MVL of the neurons (gray dotted lines) and a distribution of calculated MVLs

from shuffled data are shown. **i** Angular tuning of compass neurons and putative goal direction (GD) neurons before (left) and after (right) compass perturbation. For each neuron, the tuning of the two most directed flights are shown. Neurons are ordered according to their pfds before compass perturbation. Each neuron's firing rate (FR) is normalized against its peak firing rate. **j** Correlation of the angular tuning before and after compass perturbation ($p = 0.001$, U = 1027) and **k** heading offset variances in response to compass perturbations ($p = 0.0054$, U = 1113) for putative GD (blue, $n = 58$) and compass (green, $n = 55$) neurons (two-sided Mann–Whitney U test). Low heading offset variances indicate that the pfds were linked to the butterfly's heading. Error bars in (**j**) and (**k**) represent the interquartile range (middle line indicates the median). **l** The distributions of pfds of compass (left) and putative GD neurons (right) uniformly tiled the angular space (Rayleigh test). Source data file: datasource.xlsx Fig. 1.

larger distribution (Fig. S9b), than what would be expected from a negative phototactic behavior in monarch butterflies[48], suggesting that their orientation strategy remained a menotactic behavior.

Electric stimulation per se affected neither the orientation performance, indicated by similarly high flight precision prior to and after conditioning ($p = 0.63$, $R^2 = 0.015$, $N = 17$, two-sided paired t-test, Fig. 2d), nor the directedness of the neural tuning ($p = 0.6113$, W = −157, $n = 65$; two-sided Wilcoxon matched-pairs signed-rank test; Fig. S10).

We further excluded an effect of electric stimulation on the neural tuning through control experiments in which we showed that electric stimulation of central-complex neurons in restrained butterflies does not change the angular tuning ($p = 0.63$, W = 1136, $n = 256$, two-sided Wilcoxon matched-pairs signed-rank test, Fig. S11).

If GD neurons exist in the insect central complex, we expected that their pfds should be tightly linked to butterflies' new goal direction. Remarkably, in addition to compass neurons that showed

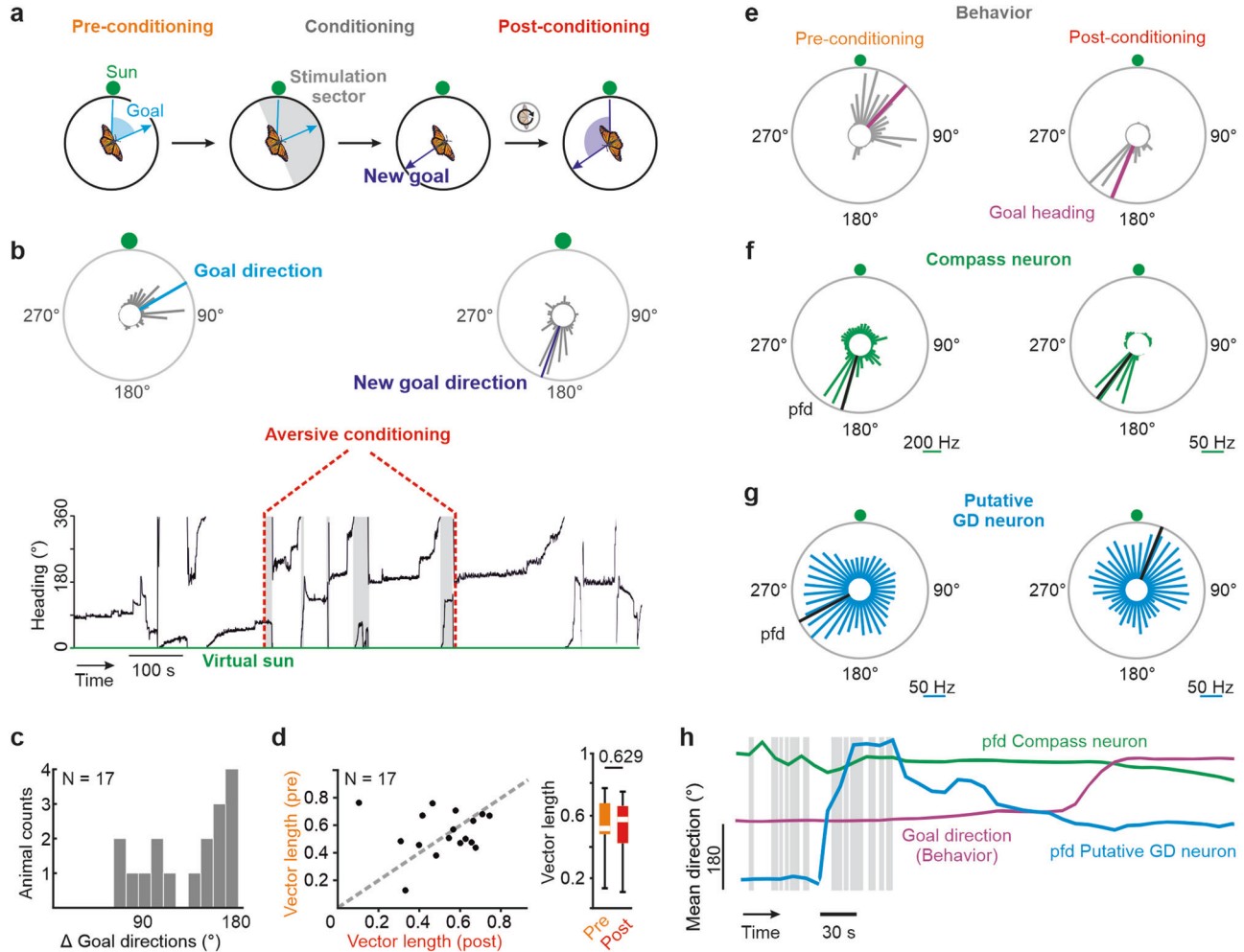

**Fig. 2 | Resetting the goal direction. a** Schematic drawing of how the butterfly's goal direction was reset. For pre-conditioning, the butterfly oriented with respect to the virtual sun. During conditioning, we applied electric shocks to the butterfly's neck whenever it headed in its initial goal direction (±90°; stimulation sector). After several electric shocks, we expected that the butterfly sets a new goal direction with respect to the virtual sun. **b** (Top) Circular histograms summarizing the heading (gray bars) of one representative butterfly before (left circular plot) and after conditioning (right circular plot). Respectively, blue and violet lines indicate the initial and new goal direction of the butterfly. (Bottom) Heading (black) plotted as a function of time. Gray boxes highlight periods of electric shocks. The virtual sun was located at 0°. **c** Changes in goal directions induced by aversive conditioning in 17 butterflies. **d** Vector lengths (flight directedness) compared between pre- and post-conditioning ($p = 0.63$, $R^2 = 0.015$, $N = 17$, two-sided paired t-test). Each dot represents the mean vector length of one butterfly. Box plots indicate median (middle line), 25th, 75th percentile (box) and 5th and 95th percentile (whiskers). **e** Circular histograms summarizing the heading of one representative butterfly before (pre-conditioning) and after (post-conditioning) conditioning. Magenta lines indicate the goal heading (mean heading). **f, g** Angular tuning of a compass neuron (green) and a putative GD neuron (blue) corresponding to the flight directions presented in (**e**). Black lines indicate the neurons' pfds. **h** Goal direction (magenta), pfds of a compass (green), and putative GD neuron (blue) plotted as a function of time. Gray boxes highlight periods of electric stimulation. The first data point on the x-axis represents the value measured during pre-conditioning. Notably, the pfd of the compass neuron was relatively invariant compared to the pfd of the putative GD neuron. The pfd of the putative GD neuron began to shift before the butterfly changed its goal direction. Source data file: datasource.xlsx Fig. 2.

invariant angular tuning (example green neuron in Fig. 2f, h), we indeed found neurons whose angular tuning changed in association with the butterflies' goal directions (example blue neuron in Fig. 2g, h).

**Angular tuning of GD neurons specifically changed with the insect's goal direction**

Similar to GD neurons in mammals, the neural activity of GD neurons in butterflies should not represent the animals' compass directions[15]. Therefore, we expected that the angular tuning of GD neurons should only change during aversive conditioning but not after compass perturbations (Fig. 3a). Consistent with this, 20 neurons (31%; HD indices <0 during compass perturbation and conditioning, Fig. S6c, d) exclusively shifted their pfds during aversive conditioning and showed invariant pfds during compass perturbations (Fig. 3b, upper heatmaps, $p = 0.012$, $W = 132$, $n = 20$, two-sided Wilcoxon matched-

pairs signed-rank test, Fig. 3c). In contrast, the angular tuning of 13 neurons (20%) changed only when we perturbed the compass (HD indices >0 during compass perturbation and conditioning; Fig. 3b, lower heatmaps), clearly showing that these are HD neurons. Comparing their spike shapes between compass perturbation and conditioning, as well as between before and after conditioning to spike shapes obtained from all recorded neurons suggested that we kept recording from the same neuron throughout the course of an experiment (two-sided Wilcoxon matched-pairs signed-rank test, $p < 10^{-5}$, $W = -3395$, Fig. S12a; $p < 10^{-5}$, $n = 82$; $W = -10276$, $n = 144$, Fig. S12b). Taken together, while the angular tuning of HD neurons was specifically affected during compass perturbations ($p = 0.01$, $t = 2.72$, two-sided unpaired t-test, Fig. 3d), the pfds of the GD neurons were only affected when the butterflies set a new goal direction ($p < 10^{-5}$, $t = 5.89$, two-sided unpaired t-test, Fig. 3e).

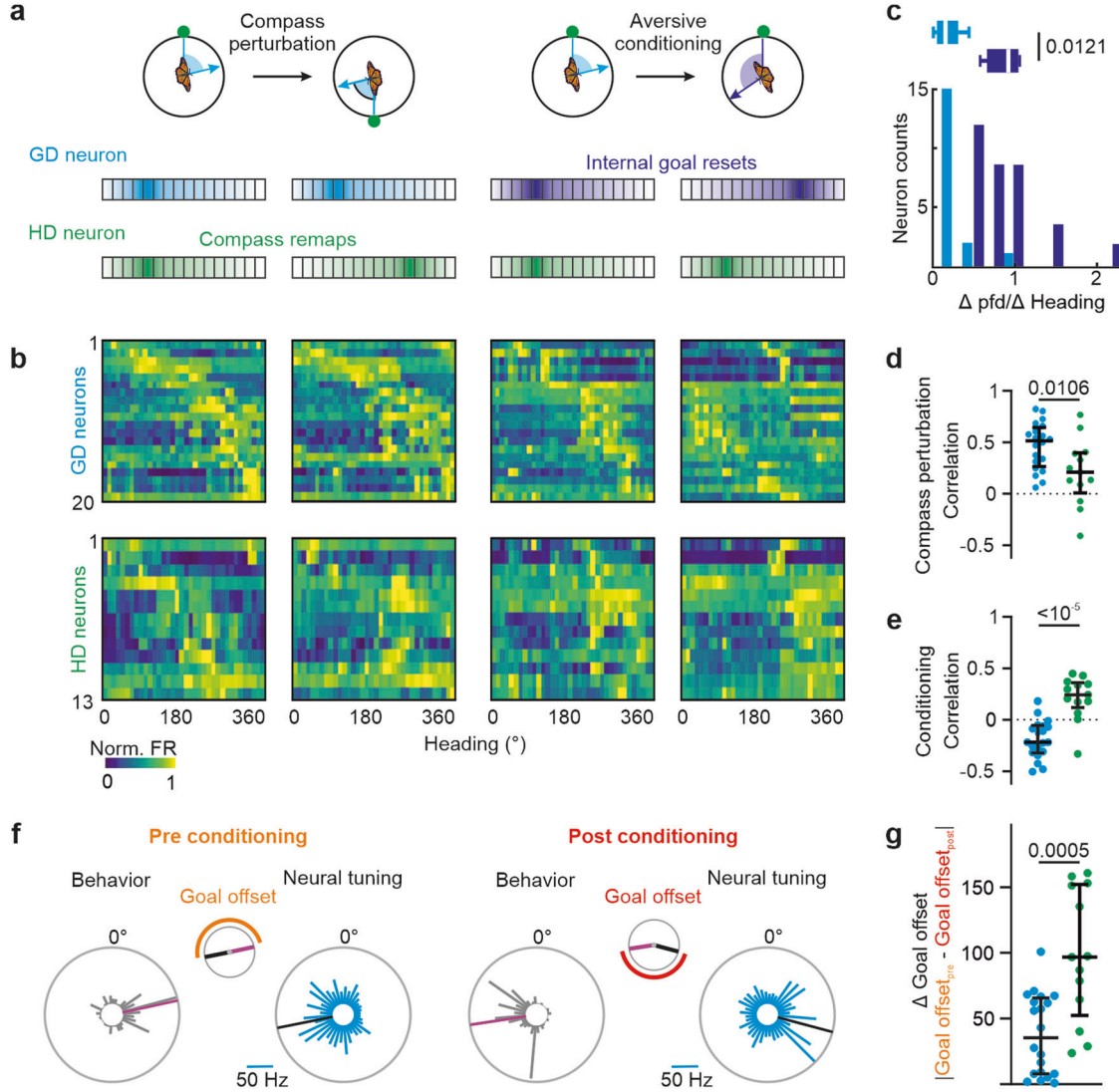

**Fig. 3 | Goal coding in monarch butterflies. a** Hypothesized tuning of a GD (blue, purple) and a HD (green) neuron in reponse to compass perturbations and aversive conditioning. The angular tuning of GD neurons was expected to change only in response to aversive conditioning, while the angular tuning of HD neurons should change only during compass perturbation. **b** Angular tuning of GD ($n$ = 20; top) and HD ($n$ = 13; bottom) neurons measured in response to compass perturbations (two left columns) and aversive conditioning (two right columns). Neurons are ordered according to their pfds before compass perturbation and conditioning. Each neuron's firing rate (FR) is normalized against its peak firing rate. **c** Ratio of changes in pfds and changes in heading for GD neurons ($n$ = 20) during compass perturbations (blue) and during conditioning (purple). Ratios close to one indicate a strong association between angular tuning and behavior. The neurons only changed their pfd when the goal direction was reset during conditioning ($p$ = 0.0121, W = 132,

$n$ = 20, two-sided Wilcoxon matched-pairs signed-rank test). Box plots indicate median (middle line), 25th, 75th percentile (box), and 5th and 95th percentile (whiskers). **d** Correlation of angular tuning before and after compass perturbations ($p$ = 0.0106, t = 2.72) and (**e**) before and after conditioning ($p < 10^{-5}$, t = 5.89) for GD (blue, $n$ = 20) and HD (green, $n$ = 13) neurons (two-sided unpaired t-test). **f** The goal offset before (orange) and after (red) conditioning describes the difference between the animal's goal direction (magenta lines) and the neuron's pfd (black lines). The change in pfd of a GD neuron was tightly associated with the change in goal direction resulting in a consistent goal offset of about 180°. **g** Differences in goal offsets before (left) and after (right) conditioning for GD (blue) and HD (green) neurons ($p$ = 0.0005 , U = 60, $n$ = 20 GD and 13 HD neurons, two-sided Mann–Whitney test). Error bars in (**d**), (**e**), and (**g**) represent the interquartile range (middle line indicates the median). Source data file: datasource.xlsx Fig. 3.

To quantify how well GD neurons encode the goal direction, we measured the goal offset representing the circular difference between the neuron's pfd and the animal's goal direction. The goal offset should be invariant throughout conditioning, i.e., a behavioral change in goal direction by -180° (gray circular plots in Fig. 3f) should be associated with a neuronal change in the pfd by -180° (blue circular plots in Fig. 3f). We found that the goal offsets were statistically smaller in GD neurons than in HD neurons demonstrating that the angular tuning of GD neurons was tightly linked to the animal's goal direction ($p < 10^{-5}$, U = 60, $n$ = 20 GD and 13 HD neurons, two-sided Mann–Whitney test; Fig. 3g). Taken together, the tight association between neural tuning and behavioral goal directions and the robust selectivity for encoding

the goal is compelling evidence that we recorded from invertebrate neurons that represented an animal's goal direction.

## GD neurons are linked to steering neurons
Central-complex models predict that GD neurons are presynaptic to steering neurons that generate pre-motor turning commands[43,44,46]. Hence, the tuning of GD and steering neurons should be closely related. As the butterflies' steering behavior changed during both compass perturbations and aversive conditioning, we expected that steering neurons should change their angular tuning in both cases (HD index >0 for compass perturbation and HD index <0 for conditioning; Figs. 4a, b, S13). During our experiments, we recorded from 19 neurons that

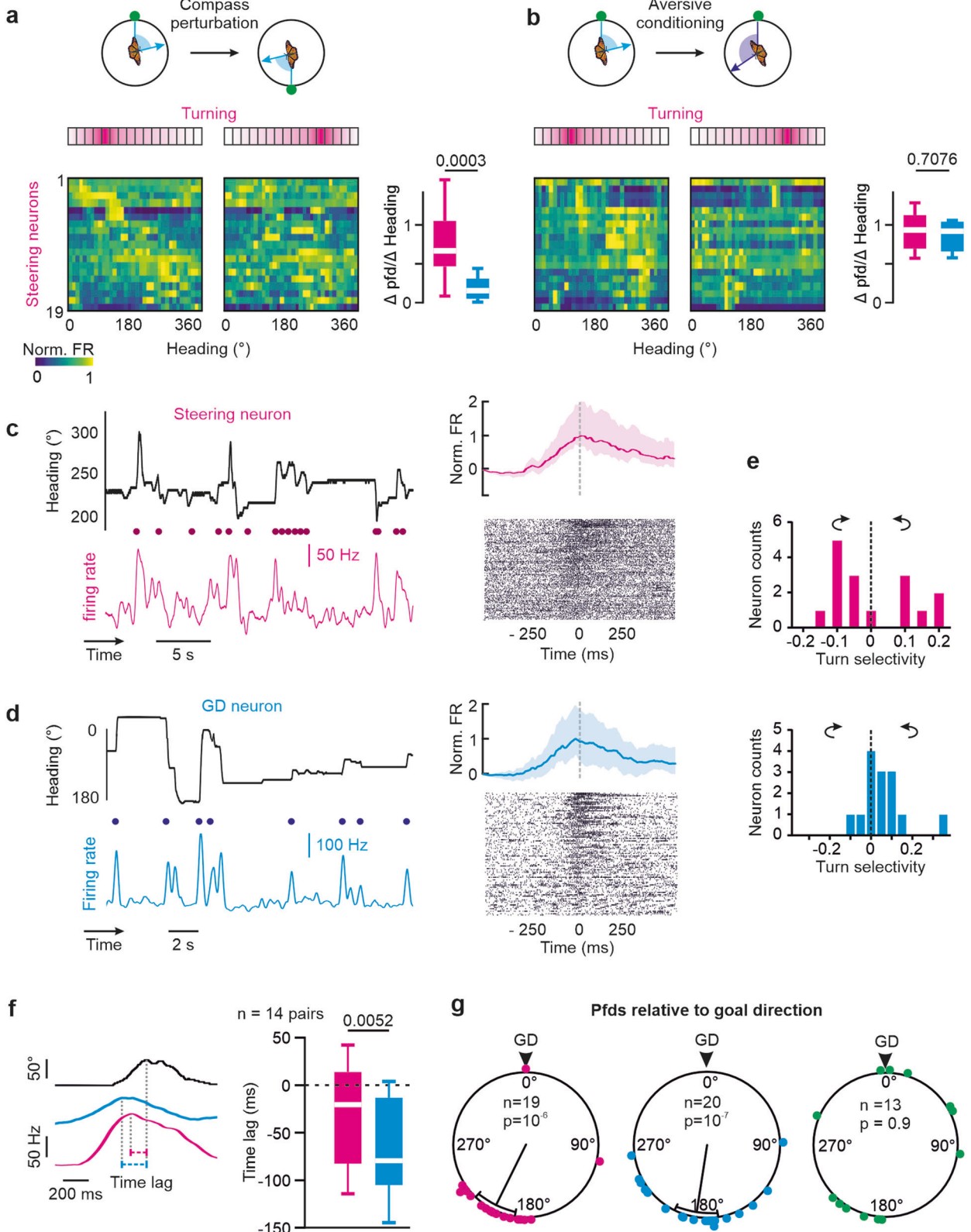

showed a tuning predicted for neurons involved in steering behavior. The angular tuning of these steering neurons was tightly linked to the butterflies' change in flight direction during compass perturbation and aversive conditioning. Typical for steering cells[42], these neurons modulated their firing rate prior to each turn of the animal (Fig. 4c). Surprisingly, the GD neurons also increased their firing rates prior to flight turns (Fig. 4d). However, while GD neurons encoded left and

right turns equally strongly, steering neurons typically exhibited a directional selectivity for one rotation direction (Fig. 4e). In addition, GD neurons that were monitored simultaneously with steering neurons in the same animal encoded turns even prior to the steering neurons ($p = 0.005$, W = −85, two-sided Wilcoxon matched-pairs signed-rank test, Fig. 4f). Our results fit well with the suggested synaptic connection between GD and steering neurons[43,44,46]. In line

**Fig. 4 | Comparison between GD and steering neurons. a, b** Change in angular tuning of steering neurons to compass perturbations (**a**) and aversive conditioning (**b**). Neurons were ordered according to their pfds before compass perturbations and conditioning. Each neuron's firing rate (FR) is normalized against its peak firing rate. Box plots show the ratio between changes in pfds and changes in heading evoked by compass perturbations ($p = 0.0003$, U = 65, **a**) and conditioning ($p = 0.0003$, U = 65, **b**) in steering (magenta, $n = 19$) and GD neurons (blue, $n = 20$) (two-sided Mann–Whitney U test). Values close to one indicate a strong association between angular tuning and behavior. **c, d** (Left) Example traces comparing heading (top) and neural firing rate (bottom) of a steering (**c**) and a GD neuron (**d**). Dots indicate time points of behavioral turns. (Right) Sliding averages (top, shaded areas represent percentile) and raster plots (bottom) summarizing the firing rates of one steering (**c**) and one GD neuron (**d**) in relation to behavioral turns at time = 0

(dashed line). **e** Directional selectivity of steering (top, $n = 16$) and GD (bottom, $n = 14$) neurons. Respectively, positive and negative values indicate stronger response preceding counterclockwise or clockwise turns. **f** (Left) Comparison of the firing rate of simultaneously recorded steering (magenta trace) and GD (blue trace) neurons during a flight turn (black trace). (Right) GD neurons modulated their firing rate prior to steering neurons ($p = 0.005$, W = −85, $n = 14$ pairs of simultaneously recorded neurons; two-sided Wilcoxon matched-pairs signed-rank test). **g** Pfds of steering (magenta), GD (blue), and HD neurons (green) relative to the butterflies' goal direction (0°). The mean pfd is shown as black line, the sectors represent the 95% confidence intervals. The $p$-values refer to the statistics from the Rayleigh test. Box plots in (**a**), (**b**), (**f**) indicate medians (middle line), 25th, 75th percentiles (boxes) and 5th and 95th percentile (whiskers). Source data file: data-source.xlsx Fig. 4.

with this proposition, pfds of GD ($p < 0.001$; V = 0.76; $n = 20$; V-test) and steering neurons ($p < 0.001$, V = 0.7, $n = 19$, V-test) were clustered in the direction opposite to the goal direction (Fig. 4g). This is contrasted by a uniform distribution of pfds in HD neurons ($p = 0.9$; Z = 0.09; $n = 13$; Rayleigh test, Fig. 4g). Our results therefore suggest that the GD neurons closely interact with steering cells and activate them whenever the butterflies substantially deviate from their desired goal direction.

## Discussion

While heading coding has been extensively studied in a variety of species[10,55–58], only little was known about goal direction coding[51]. We here described the coding of GD neurons in the insect brain. The angular tuning of these neurons changed when the butterfly's goal direction was reset (Fig. 2). More importantly, this change was tightly associated with the change in goal direction (Fig. 3g). In contrast to this, compass perturbations did not affect the angular tuning in the very same neurons (Fig. 3). This specific coding of the goal direction is well in line with very recent findings from the *Drosophila* brain[59] and confirms that the insect brain houses GD neurons similar to the ones previously discovered in the mammalian hippocampus[15]. Consistent with our results, mammalian GD neurons did not represent heading information but were specifically tuned to the spatial goal. Some neurons in the mammalian hippocampus were additionally tuned to the goal distance[15,21]. Processing goal distance information is particularly important for vector navigation[26,60] and for homing[61,62]. Recent results in the brain of a variety of insects show that the central complex additionally processes distance information[25,63], making it highly likely that the insect GD neurons are also tuned to distance in the context of path-integration[64–66] or prey detection[63]. Whether distance coding has any behavioral relevance in non-migratory or migratory monarch butterflies remains unclear. Like migratory birds, migratory monarch butterflies rely most likely on a stop signal rather than distance information to localize their migratory goal[67]. Thus, it would not be surprising if the here described GD neurons in monarch butterflies do not encode distance information and purely encode directional information for short-distance dispersal (non-migratory) or long-distance migration (migratory).

The monarch butterfly GD neurons, like the bat GD neurons[15], are biased in their tuning. In contrast to the bat GD neurons[15] and goal-modulated place cells in rodents[16,21], which both tend to increase their firing rate when the animal faces its goal, the insect GD neurons respond maximally when the animal heads in the direction opposite to the goal direction. According to a recent model on the insect navigation circuit, a maximum activity of the GD neurons when the animal heads in the anti-goal direction helps to reliably encode the insects goal direction[46].

Recent studies predict that GD neurons innervate the fan-shaped body of the central complex[25,26,43,46,59] which fits well with our recording site (Fig. S4). However, as most central-complex neurons, including HD and steering neurons, also project through the fan-shaped body[1,45,68],

our technique does not allow us to conclude where exactly these neurons are localized in the central complex. Interestingly, a recent study confirms the existence of GD neurons, termed FC2 neurons, in the *Drosophila* fan-shaped body[59]. If our GD neurons are similar cells as the FC2 neurons, remains to be determined in the future.

Due to the highly conserved nature of the central complex[1,36,37,69], our results give deep insights into the general coding of goal-directed orientation in insects. Our recordings were obtained from non-migratory monarch butterflies that are closely related to the population of migratory monarch butterflies but lost their ability to migrate[70,71]. Thus, in contrast to the single (southward) goal-direction set by the population of migratory monarch butterflies, the non-migratory butterflies maintain any possible goal direction with respect to a virtual sun (menotactic orientation)[48,53]. Because non-migratory monarch butterflies demonstrate individual-specific goal directions[48,53], we reasoned that their goal directions can be experimentally controlled, which is ideal to investigate the neural coding of goal directions. However, as non-migratory captive-reared monarch butterflies differ behaviorally[70,71], morphologically[72–74], and physiologically[73] from migratory monarch butterflies[28,70,75–78], ideas on how the migration behavior is encoded in the monarch butterfly brain should be read with cautious, here.

Despite the differences found between migratory and non-migratory monarch butterfly populations, the anatomy of the central-complex network in the monarch brain can be expected to be highly similar, even down to single sun compass neurons[33,79]. Differences in the coding of goal directions between the migratory and non-migratory monarch butterflies had been discussed to underly synaptic modifications of the same neurons[25,37]. Such synaptic modifications may explain volumetric differences of some brain regions in migratory and non-migratory monarch butterflies[79]. Based on our results from non-migratory monarch butterflies, we predict how the tuning of the same central-complex neurons could be modified to encode long-distance migration in migratory monarch butterflies: Like in all other insects, the monarch butterfly fan-shaped body is compartmentalized into 16 columns[79]. We predict that a population of GD neurons, homologous to the ones described in this study, represents the migratory southward direction within the columns of the fan-shaped body[1,2,59], similar to how the HD network represents a compass of heading directions across the columns of the ellipsoid body of the central complex[3,4,57]. By changing the compass polarity through sun displacements, we might have induced a translocation of the HD representation in the butterflies' ellipsoid body, as it has also been demonstrated in *Drosophila*[3]. Contrastingly, the GD representation was unaffected by compass perturbations[59]. Resetting the goal direction through aversive conditioning, however, might have induced a translocation of the GD representation across the columns of the fan-shaped body. We predict that a similar translocation of the GD representation could transform the butterfly's southward direction into a northward one in migratory monarch butterflies before departing for their remigration in spring[80]. As migratory monarch

butterflies substantially differ from non-migratory monarch butterflies in terms of endocrinology[68], reproduction[29], longevity[68], metabolism[73], and morphology[74,81–83], it is fundamental to test whether these differences may also affect the neural coding of migration in monarch butterflies in the future.

The representation of the current heading in the ellipsoid body and the goal direction in the fan-shaped body are thought to be compared by downstream steering neurons[25,26,46]. Recent findings in *Drosophila* suggest that this comparison is carried out by two different neuron types, termed PFL3 and PFL2[43,44,59]. PFL3 neurons are active and induce steering commands whenever the insect deviates from its intended goal direction, whereas PFL2 neurons are primarily active when the insect heads into the anti-goal direction[44]. This is well in line with the monarch butterfly steering neurons described here and suggests that these neurons transfer motor commands to descending neurons whenever the butterfly is flying into the anti-goal direction. In migratory monarch butterflies, this mechanism would allow the animal to effectively steer back to its southward goal direction when it strays off course.

Taken together, the orientation network of insects consists of different neuron types processing the current heading direction and goal direction, generating steering commands whenever the butterfly deviates from its course. In this study, we physiologically described GD neurons in the insect brain that, similar to GD cells reported in bats, represent the goal direction in an egocentric frame of reference.[15] This emphasizes the evolutionary origin of goal coding in animal navigation.

## Methods

### Animals

Monarch butterflies (*Danaus plexippus*) were ordered as pupae from Costa Rica Entomological Supply (butterflyfarm.co.cr) and kept in an incubator (HPP 110 and HPP 749, Memmert GmbH + Co. KG, Schwabach, Germany) at 25 °C, 80% relative humidity and 12:12 light/dark-cycle conditions. After eclosion, adult butterflies were transferred into another incubator (I-30VL, Percival Scientific, Perry, IA, USA) at 25 °C and 12:12 light/dark condition. Adults had access to 15% sucrose solution ad libitum. We performed our recordings from both sexes without any intended bias as no sexual difference in directional coding was expected.

### Behavioral monitoring

A magnet (diameter = 3 mm; magnetic force = 4N Supermagnete, Webcraft GmbH, Gottmadingen, Germany) was dorsally attached with dental wax (Article: 54895 Omnident, Rodgau Nieder-Roden, Germany) to the thorax of 32 butterflies of both sexes. A second magnet at the end of a tungsten rod was used to connect the butterfly dorsally to an optical encoder (E4T miniature Optical Kit Encoder, US Digital, Vancouver, WA, USA) which measured the animal's heading direction at a sampling rate of 100 Hz and at an angular resolution of 3°. Encoder signals were digitized (USB4 Encoder Data Acquisition USB Device, US Digital, Vancouver, WA, USA) and visualized in the US Digital software (USB1, USB4: US Digital, Vancouver, WA, USA). The optical encoder was vertically attached to a micro linear actuator (L12-R 50 mm 50:1 6 Volts, Actuonix Motion Devices, Saanichton, BC, Canada) that allowed us to control the butterfly's suspension height using an Arduino MEGA 2560. The tethered butterfly could steer along any azimuth while being suspended at the center of a custom-built flight arena. The arena had an inner diameter of 32 cm and a height of 12 cm, and its upper inner circumference was equipped with 144 RGB-LEDs (Adafruit NeoPixel, Adafruit Industries, New York, New York, USA). The LED strip was mounted at an elevation of ~30° relative to the butterfly. One of these LEDs provided a single green light spot that served as a virtual sun stimulus ($1.74 \times 10^{13}$ photons/cm$^2$/s and 1.2° angular extent at the butterfly's eyes, as measured at the center of the arena). The angular position of the virtual sun was controlled by the Arduino MEGA 2560.

## Neural recordings

For neural recordings, one ($N = 9$) or three tetrodes ($N = 23$) were implanted in the butterfly central-complex. Each tetrode comprised a bundle of four 18 cm long and 12.5 μm thin copper wires (P155, Elektrisola, Reichshof-Eckenhagen, Germany) that were waxed tightly together. In experiments in which only one tetrode was implanted, the tetrode consisted of five copper wires (four recording and one differential wire). Tetrodes were carefully threaded through two Pebax® tubes (each 2–4 cm in length; 0.026' inner diameter; Zeus Inc, Orangeburg, SC, USA) that served as anchoring points to reversibly mount the tetrodes to a glass capillary. An additional copper wire served as grounding electrode and was immersed into the head capsule close to the butterfly's neck. For aversive conditioning ($N = 17$), two stimulation copper wires (resistance -10 MΩ) were waxed to the grounding electrode. All copper wires were soldered to gold pins and attached to an electrode interface board (EIB-18; Neuralynx Inc., Bozeman, MT, USA). In experiments in which three tetrodes were used, the tetrodes were fanned to maximally span 200–250 μm along the horizontal axis. Before each experiment, electrode resistances were measured with a nanoZ (Multi Channel Systems MCS GmbH, Reutlingen, Germany) and the electrode tips plated (Elektrolyt Gold solution, Conrad Electronic SE, Hirschau, Germany) to reduce the resistance of each electrode to -0.1–1 MΩ. Tetrodes were reused for multiple experiments, after carefully trimming the tips and replating to the desired resistance.

Prior to obtaining neural signals of central-complex neurons, a monarch butterfly was horizontally restrained on a magnetic holder. To minimize movement artifacts during the recordings, the head was waxed to the thorax. The head capsule was opened dorsally and fat and trachea covering the brain surface were removed. To gain access to the central complex, the neural sheath on the dorsal brain surface was carefully removed using fine tweezers. The electrode bundle containing the grounding and the stimulation wires were inserted posteriorly in the head capsule, close to the butterfly's neck. Tetrode tips were immersed in ALEXA 647 fluorophore coupled Hydrazide (A20502 diluted in 0.5 M KCl, Thermo Fisher Scientific GmbH, Dreieich, Germany) to quantify the tetrode position after each experiment. Recording tetrodes were then inserted into the brain, once per experiment. Tetrodes together with the glass capillary were attached to an electrode holder (M3301EH; WPI, Sarasota, FL, USA) and their positions controlled via a micromanipulator (Sensapex, Oulu, Finland). After adjusting the tetrode position along x- and y-axes, hemolymph fluid covering the brain was temporarily removed and the tetrodes were carefully moved along the z-axis to reach the central complex. While moving along the z-axis, band-pass filtered (600–6000 Hz) neural signals were measured at a sampling frequency of 30 kHz. Neural signals were sent from the EIB-18 via an adapter board (ADPT-DUAL-HS-DRS; Neuralynx Inc., Bozeman, MT, USA) to a Neuralynx recording system (DL 4SX 32ch System, Neuralynx Inc., Bozeman, MT, USA). The neural activity was monitored using the software Cheetah (Neuralynx Inc., Bozeman, MT, USA). For setting a differential configuration, one electrode of the neighboring tetrode was set as a reference for the recording tetrode in the software. This means that the neural signals of each tetrode were referenced against the neural signal of an electrode of the neighboring tetrode. In cases in which only one tetrode was implanted, one of the five copper wires of the recording tetrode served as a reference. To find visually sensitive neurons, the virtual sun was occasionally revolved clockwise and counterclockwise at an angular velocity of 60 deg/s around the insect's head and the neural responses were visually quantified. After finding visually sensitive neurons at depths between 150–450 μm, the tetrode and the grounding wire were held in place by adding a two-component silicone elastomer (Kwik-Sil, WPI, Sarasota, FL, USA). After the Kwik-Sil hardened (-1 h), the butterfly was carefully unrestrained and connected via the magnet to the end of the tungsten rod that was connected to the optical encoder. The tetrodes were carefully removed from the

glass capillary and attached to a Pebax® tube that was orthogonally oriented to the tungsten rod. To avoid wrapping the tetrode wires around the tungsten rod while the butterflies steered, the animals' angular movements were restricted to 358°. To synchronize behavioral and neural recordings offline in Spike2 (version 9.0 Cambridge Electronic Devices, Cambridge, UK), a trigger signal was sent from the USB4 encoder via an ATLAS analog isolator (Neuralynx Inc., Bozeman, MT, USA) and the adapter board to the Neuralynx recording system at the onset of the behavioral recording. To temporally align stimulus presentations with the recorded neural activity, an analog output of the Arduino was sent via the ATLAS analog isolator to the Neuralynx recording system.

### Visualization of electrode tracks

After the neural recordings, the brain was dissected out of the head and fixated overnight in 4% formaldehyde at 4 °C. The brain was then transferred into sodium-phosphate buffer and rinsed for 2 × 20 min in 0.1 M phosphate buffered saline (PBS) and 3 × 20 min in PBS with 0.3% Triton-X. The brain was dehydrated with an ascending ethanol series (30–100%, 15 min each) and immersed with a 1:1 ethanol-methyl-salicylate solution for 15 min, followed by a clearing step in methyl-salicylate for at least 1 h. It was then embedded in Permount (Fisher Scientific GmbH, Schwerte, Germany) between two cover slips and scanned with a confocal microscope (Leica TCS SP2, Wetzlar, Germany) using a 20x water immersion objective (HC PL APO CS2 20x/0.75 IMM, Leica, Wetzlar, Germany). To visualize the tetrode position, we reconstructed the tetrode tracks in 3D using the software Amira 5.3.3 (ThermoFisher, Germany). To compare the tetrode positions from different experiments, we registered the tetrode position into the monarch butterfly standard central complex[79]. We used an affine (12-degrees of freedom), followed by an elastic registration to transfer the neuropils of the individual central complexes into the corresponding neuropils of the standard central complex. The registration and deformation parameters were then applied to the tetrode reconstruction to visualize the tetrodes in one frame of reference.

### Spike sorting and spike shape analysis

Neural recordings were spike sorted with the tetrode configuration implemented in Spike2 (version 9.00, Cambridge Electronic Devices, Cambridge, UK). We used four spike detection thresholds (two upper and two lower thresholds). The highest and lowest thresholds were set to avoid misclassifications of large voltage deflections occasionally arising from flight movements as spikes. The time window for template detection was set to 1.6 ms. After spike-sorting, a principal component analysis (PCA) was used to evaluate and to redefine spike clusters. Spike2 channels were exported as down-sampled Matlab files (3 kHz) and the remaining analysis was done with custom written scripts in MATLAB (Version R2021a, MathWorks, Natick, MA, USA). To analyze the spike shapes, the WaveMark channels containing the spike-waveforms were additionally exported as non-down-sampled Matlab files (30 kHz). For each neuron, spike-waveforms averaged from the first half of the experiment (compass perturbation) were correlated with the averaged spike-waveforms of the second half of the experiment (aversive conditioning) and statistically compared with the averaged spike-waveforms of the remaining neurons (two-sided Wilcoxon matched-pairs signed-rank test). This quantification allowed us to statistically test whether neural recordings were stable throughout the experiment and assessed the quality of our spike-sorting analysis.

### Quantifying behavior and neural tuning

For behavioral analysis, we computed circular histograms by adding each data point of the optical encoder to the corresponding 10-degree heading bin. The animal's preferred heading, represented by the mean vector, was computed with the CircStat toolbox in MATLAB. The flight directedness (r) was described with the mean vector strength which ranged between 0 (non-directed) to 1 (highly directed). Distributions of preferred headings of all animals were tested for uniformity with a Rayleigh test and visualized in Oriana (Version 4.01, Kovach Computing Services, Anglesey, Wales, UK).

To characterize the spatial coding in the tethered butterflies, the tested animals should have ideally explored all possible heading directions uniformly. However, when the goal is to maintain a directed course and the animal's heading is thus biased towards the goal direction, as it was the case in the current study, a uniform heading representation is experimentally impossible[8]. To minimize the effect of a biased heading representation on the angular tuning, we quantified the angular tuning based on the neurons' mean firing rates instead of accumulating single spike events. Moreover, during the experiments, the butterflies had to explore each possible heading direction to allow us to calculate the spatial tuning. Despite the biased heading representation, changes in the mean heading did not affect the angular tuning of many central-complex neurons. For example, perturbations of the butterfly's compass eliciting a substantial change in the mean heading (Fig. 1d) did not affect the angular tuning of putative GD neurons (blue neuron in Fig. 1h). Similarly, the angular tuning of HD neurons was constant (Fig. 2g) despite substantial changes in the butterfly's mean heading elicited by aversive conditioning (Fig. 2e). We therefore reasoned that the sampling bias in heading directions did not affect the angular tuning of central-complex neurons.

Directional coding of each neuron was quantified from circular plots. For each behavioral condition, i.e., compass perturbation, pre-, post-conditioning, a circular plot was calculated that reflects the mean firing rate at different heading directions (10-degree bins). Circular statistics were then computed using the CircStat toolbox for MATLAB or in Oriana (Version 4.01, Kovach Computing Services, Anglesey, Wales, UK). First, angular sensitivity was determined by testing whether the mean firing rate deviated from a uniform distribution (Rayleigh test; significance level $\alpha = 0.05$). If this was the case, we calculated the mean vector, or preferred firing direction (pfd), of a neuron.

### Dark experiments

To focus on neurons that showed an internal representation (GD neurons) or are tuned to idiothetic cues, i.e., in the absence of visual signals (HD neurons), we allowed the butterflies to orient on a Lab Jack prior to flight (Compact Lab Jack, Inc, Newton, New Jersey, USA). After the butterflies could steer in the presence of a virtual sun for a couple of minutes, we turned off the virtual sun and measured neural signals from the butterfly orienting in darkness. 113 out of 147 recorded neurons preserved their angular sensitivity when the butterflies were orienting in darkness and all subsequent neural analysis were based on these 113 neurons (Rayleigh test: significance level $\alpha = 0.05$).

### Compass perturbation

To perturb the butterfly compass, we performed a similar experiment as the one performed in *Drosophila*[3]. However, instead of a vertical bar, we used the virtual sun as reference point of the insect compass. In the presence of the virtual sun, the butterfly flew for 9 min, and we changed the angular position of the sun every 90 s. In 15 experiments we changed the sun position in decreasing steps of 180°, 90°, 45°, 23°, and 15°. For the remaining 17 experiments, we exclusively changed the sun position in relatively large steps of 90° (3 times/experiment) or 180° (2 times/experiment). Preferred headings were measured every 90 s.

### Measuring tuning directedness (mean vector length)

Tuning directedness of the neurons was quantified by calculating the mean vector length (MVL) of angular tuning (Rayleigh statistics). The MVL of each neuron was statistically compared to a distribution of

MVLs generated by a permutation (1,000,000 repetitions). The permutation shuffled the 10-degree bins of the circular plots and computed a MVL after each shuffle. MVL of the real data were significantly longer than the MVLs computed from the shuffled data from each neuron ($p < 0.05$; Fig. 1g). In addition to the neuron-wise comparison of MVL and MVLs from the shuffled data, we statistically compared the measured MVLs from all the neurons with the MVLs after shuffling the data from all 113 neurons ($p < 10^{-5}$, W = −10878, $n = 147$, two-sided Wilcoxon matched-pairs signed-rank test).

### Functional classification of neurons (HD-index)

Neural data were considered from three periods, in which the animals showed the highest flight directedness (vector length). Neurons were categorized regarding their changes in pfds in response to sun displacements. To categorize if the pfds of neurons changed with the animal's heading, we calculated an HD index. First, we calculated the heading offset, which represents the angular difference between pfd and behavioral heading directions. We then computed the circular variance of these heading offsets (CVH) for each neuron. As neurons linked to the animal's heading should change in accordance with the behavior, we expected that their CVH should be relatively low (Fig. S6). In contrast, neurons that are not linked to the animal's heading, such as GD neurons, should reveal relatively high CVH values. In addition, we computed the circular variance of all pfds (CV). The CV should be relatively low for neurons with an invariant tuning, such as GD neurons and high for neurons with a variant tuning, such as compass neurons (Fig. S6). The HD index for each neuron was then calculated with the following equation:

$$\text{HD index} = \frac{(CV - CVH)}{(CV + CVH)} \qquad (1)$$

An HD index >0 indicates that the neural tuning can better be explained by a correlation with the animal's heading (putative HD & steering neurons, $n = 55$), while an HD index <0 indicates that neural tuning was unaffected by the animal's heading (putative GD neurons, $n = 58$). In addition, we correlated the binned neural response (10-degree bin size) measured prior to sun displacement with the one measured after displacement.

### Resetting the internal goal direction through aversive conditioning

To reset the butterfly's internal goal direction without perturbing the compass system, we coupled the initial goal direction (±90°) with electric shocks (U = 5 V; I = 0.5 μA). Prior to aversive conditioning (pre-conditioning), the initial goal direction was visually determined by the experimenter while the butterfly oriented with respect to a static virtual sun. Depending on the butterfly's motivation to keep a consistent heading, this could take several minutes. To reset the goal direction by a significant amount, the butterfly received electric shocks whenever it flew in a sector containing the initial goal direction ±90° (aversive conditioning). Electric shocks were controlled in the US Digital software (USB1, USB4: US Digital, Vancouver, WA, USA) that sent a signal from one of the USB4 output channels (USB4 Encoder Data Acquisition USB Device, US Digital, Vancouver, WA, USA) to the stimulus lines at the Neuralynx adapter board (ADPT-DUAL-HS-DRS; Neuralynx Inc., Bozeman, MT, USA). In parallel, the time course of stimulation was monitored by sending a digital signal from the USB4 to the Neuralynx system via the ATLAS analog isolator (Neuralynx Inc., Bozeman, MT, USA). Aversive conditioning took several minutes, depending on the butterfly's performance. After aversive conditioning, the butterfly was allowed to freely steer with respect to the virtual sun for several minutes (post-conditioning). Note that the virtual sun's azimuth was constant throughout the conditioning to avoid any compass perturbations. Heatmaps comparing the heading directions prior to

and after conditioning were computed by normalizing the circular histograms containing the headings against the maximum bin. To roughly compare changes of preferred headings (behavior) and pfds (neurons) over time, we moved a sliding window in 10 s steps from the beginning of the aversive conditioning to the end of the experiment. To compute a preferred heading/pfd for each time window, it was necessary that the butterfly headed in each direction. Therefore, time window sizes were relatively large (mean/std: 262/78 s) and were set from the beginning of pre-conditioning to the time point of the first electric shock. Temporal dynamics of pfds were only measured for the purposes of visualization. For quantitative analysis, we compared the angular tuning measured by circular plots between pre- and post-conditioning. 65 neurons that were categorized as putative HD/steering and GD neurons from the sun displacements were further categorized by calculating an HD index based on the conditioning experiment. In contrast to the compass perturbation, resetting the goal direction should selectively affect GD and steering neurons, while HD neurons should demonstrate a consistent tuning. By determining two HD indices (one measured in response to compass perturbation and a second one in response to conditioning), we categorized four groups of neurons. (i) HD > 0 in both cases represent HD neurons ($n = 13$). (ii) HD > 0 for compass perturbation but HD < 0 for conditioning represent steering neurons ($n = 19$). (iii) HD < 0 in both cases represent GD neurons ($n = 20$). (iv) HD < 0; HD > 0; It should be noted that we also obtained recordings from neurons with invariant pfds during both sun displacement and conditioning ($n = 13$). As we could not explicitly test and interpret their coding, we decided to not consider them further in this study.

### Electric stimulation experiments in restrained butterflies

In control experiments aiming to test whether electric stimulation affects neural tuning, two stimulation copper wires (resistance: ~1 MΩ) were mounted on a single tetrode and inserted into the central complex of a restrained butterfly. The proximity of stimulation electrodes to the recording site allows one to undeniably test whether electric stimulation affects neural tuning in the central complex. For visual stimulation, the virtual sun was revolved clockwise and counter-clockwise at an angular velocity of 60°/s around the butterfly. Electric stimulations were applied as pulses (1 ms) and repeated at 20 and 40 Hz with an electric current of 0.5–5 μA. Note that we even tested higher currents than the one used for aversive conditioning. Angular tuning, including pfds of 256 neurons were compared between pre and post stimulation (two-sided Wilcoxon matched-pairs signed-rank test).

### Testing for coding of turning behavior

To test for coding of flight turns, we determined the time points when the animal's heading changed by more than 9°. We set 9° as the turn threshold because the encoder's angular resolution was 3° and deviations of ±3° could represent variations in flight direction which may not represent substantial flight turns. In 1 s time windows, we examined the neurons' firing rate prior to (−500 ms) and after (+500 ms) the flight turns. Sliding averages of the neural activity were generated by applying a low-pass filter to the inter spike-intervals of the neurons. The neural activity in each time window was normalized to the firing rate 500 ms prior to the turn. Neural activity in time windows in which no flight turn occurred were considered as controls and statistically compared with the neural activity recorded during turns. Neurons were categorized as coding for flight turns if (i) modulations in the neural activity during flight turns were higher/lower than the modulations in neural activity during control (two-sided Wilcoxon p-test <0.05) and (ii) if modulations in the neural activity during flight turns fitted a Gaussian distribution (>0.7). Time lag between the peak firing rate and the maximum angular velocity (behavior) were computed by cross correlating the neural activity with the angular velocity. Negative time lags indicate that the neural activity changes prior to angular

turns and vice versa. Neurons coding for flight turns were tested whether clockwise or counterclockwise turns elicited responses of different strengths by calculating a "turn selectivity". Hereby, the peak firing rate in response to clockwise (CW) and counterclockwise (CCW) rotations were compared and weighted by the following equation:

$$\text{Turn selectivity} = \frac{(\max \text{CCW} - \max \text{CW})}{(\max \text{CCW} + \max \text{CW})} \qquad (2)$$

## Statistics

Circular statistics were performed in MATLAB and Oriana (Version 4.01, Kovach Computing Services, Anglesey, Wales, UK). Linear statistics were computed in GraphPad Prism 9 (GraphPad Software, San Diego, CA, USA). Sample sizes were not statistically pre-determined. Data distributions were tested for normality with a Shapiro–Wilk test. Normally distributed data were further analyzed with parametric statistical tests, while non-normally distributed data were tested with non-parametric tests. A Rayleigh test testing for uniformity of circular data was used to examine whether the flights were biased toward any direction. To statistically compare the angular tuning measured prior to and after compass perturbation across compass and putative GD neurons, we compared the correlation values obtained by correlating the angular tuning prior to sun displacement with the one measured after sun displacement with a two-sided unpaired t-test (Fig. 1j). Heading offsets and circular variances of pfds were statistically compared with a two-sided Mann–Whitney U test (Fig. 1k and Fig. S7c). Variations in spike rate across compass and putative GD neurons were also compared with a two-sided Mann–Whitney U test (Fig. S8). Changes in goal directions induced by aversive conditioning were statistically tested by comparing the distribution of GDs before conditioning (pre-conditioning) with the ones after conditioning (post conditioning) using a Mardia-Watson-Wheeler test (Fig. 2c). Flight directedness and directedness of neural tuning prior to and after conditioning was compared with a two-sided paired t-test (Fig. S10) and a two-sided Wilcoxon matched-pairs signed-rank test (Fig. 2d), respectively. To compare the tuning stability prior to compass perturbation and aversive conditioning with the one measured after compass perturbation and aversive conditioning, we statistically compared the correlation values obtained by comparing the angular tunings with a two-sided ordinary one-way ANOVA across different neuron types, i.e., HD, GD, and steering neurons (Fig. S13b). Note when comparing between two neuron types, we used a two-sided Mann–Whitney U test (Fig. 3d, e). A two-sided Mann–Whitney U test was used to statistically compare the changes in pfds induced by aversive conditioning and compass perturbations in GD neurons (Fig. 3c) and when comparing pfd changes induced by compass perturbation and aversive conditioning between GD and steering neurons (Fig. 4a, b). Time lags of turn coding were statistically compared across steering and GD neurons with a two-sided Mann–Whitney U test (Fig. 4f). Hereby, only pairs ($n = 14$ pairs) of simultaneously recorded steering and GD neurons were considered because a comparison of time lags across different experiments were unprecise due to the relatively low sampling rate of the optical encoder. The consistency of goal offsets for putative GD and HD neurons across the conditioning was statistically compared with a two-sided Mann–Whitney U test (Fig. 3g). Goal offset stability was statistically compared between GD, HD, and steering neurons (two-sided Kruskal–Wallis test; one-way ANOVA; Fig. S13c). Using a Rayleigh test, we examined whether pfds of HD neurons were uniformly distributed and a V-test (expected 180°) allowed us to demonstrate that pfds of GD and steering neurons were clustered at 180° (Fig. 4g).

Data collection and analysis were not conducted blindly to the conditions of the experiments. For neural recordings, stimulus presentation was pseudorandomized. We excluded 34 of the 147 recorded neurons, because of the lack of angular tuning when the butterflies oriented in darkness on a platform prior to flight (Rayleigh test: $p > 0.05$; see also Fig. S5).

## Reporting summary

Further information on research design is available in the Nature Portfolio Reporting Summary linked to this article.

## Data availability

Data generated in this study have been deposited in the WueData database[84] under accession code https://doi.org/10.58160/92. Source data are provided with this paper.

## Code availability

Matlab scripts programmed for this study have been deposited in the WueData database[84] under accession code https://doi.org/10.58160/92.

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

## Acknowledgements
We thank Marie Dacke, Lena van Giesen, Eric Warrant, Emily Baird, Kang Nian Yap, Alice Chou, and Stanley Heinze for their helpful comments on our manuscript. We thank Martin Strube-Bloss, Myriam Franzke, Keram Pfeiffer, Wolfgang Rößler, and Konrad Öchsner for their technical support. In addition, we thank Sergio Siles (butterflyfarm.co.cr) and Marie Gerlinde Blaese for providing us with monarch butterfly pupae. This work was funded by the Emmy Noether program of the German Research Foundation (Grant number: EL784/1-1; granted to B.e.J.). This publication was supported by the Open Access Publication Fund of the University of Würzburg.

## Author contributions
Conceptualization (M.J.B., B.e.J.), methodology and formal analysis (M.J.B.), investigation (M.J.B., C.K.), visualization (M.J.B., C.K., B.e.J.), funding acquisition (B.e.J.), project administration and supervision (B.e.J.), writing—original draft (M.J.B., B.e.J.), writing—review & editing (M.J.B., C.K., B.e.J.).

## Funding

## Competing interests
The authors declare no competing interests.
