## [Peer Review File · Nature Communications]

Neural representation of goal direction in the monarch butterfly brainREVIEWER COMMENTS

Reviewer #1 (Remarks to the Author):

The manuscript by Beetz and colleagues uses multi-electrode recordings from the central complex of the monarch butterfly to describe 3 classes of neurons: heading direction, goal direction, and steering neurons. The classes are delineated through a clever series of behavioral perturbations in which either a heading cue is rotated or the goal direction of the animal is altered through negative reinforcement. Overall the results are exciting and speak to an emerging understanding on the field of how navigational goals are encoded and read out to meet the behavioral needs of diverse animal species. My main concern is that the division in 3 neuronal classes may be too simplified; the authors might do a better job of illustrating the range of cell types and responses in their data set. As well, the introduction and discussion over-emphasize the novelty of the work at the expense of setting the findings appropriately in the larger context of understanding navigational systems. Finally, some aspects of the figures are not well explained in the legends. Overall I think this is a valuable contribution to the field that can be improved with some additional analysis and editing of the text.

Major comments:

1) Division of recorded cells into HD and GD neurons.

The manuscript is framed around the discovery and division of CX neurons into heading direction (compass) and goal direction neurons. Figure 1G shows examples and H-J quantify differences in these populations. However, how cells were classified into these two types is not totally obvious. I would expect the heading direction neurons to shift their tuning curves by 180° with the compass perturbation shown in H, but that does not seem to be what happens from the heat-maps in H (top right plot). For the quantifications in I and J I can't tell how seriously to take these comparisons because I am not sure what the criteria were for dividing cells into the two categories. Figure S6C suggests that what look like a smooth gaussian distribution of properties has been divided in half to form two categories of neurons. Overall I think the authors need to do a better job of (1) illustrating the full diversity of neuronal response types, (2) defining explicit criteria by which they have categorized neurons as HD or GD, and (3) explaining their thinking about the diversity of cell and response types and how they might function in navigation. Recent results from other insect species I think suggest that there are likely many different cell types in the CX with a diversity of response properties perhaps play different roles in different behaviors. I think a better statistical description of the distribution of responses seen in the butterfly would contribute to our understanding of this diversity and its potential roles in butterfly navigation behavior.

2) Framing of the manuscript within the context of recent literature.

The authors would like to draw attention to the novelty of their finding of goal direction neurons, but I think in doing so they mis-represent the current state of the literature. For example, the abstract states that the coding of goal direction is “completely unexplored insects” which is not true. Multiple computational studies have proposed the existence and nature of goal signals, and experimental evidence for goal signals in insects has been shown recently through optogenetic activation in flies as well as recordings in more than one study. Many novelty statements in the Abstract, first paragraph of the Introduction, and Discussion should be toned down and set in a more appropriate context of other studies (see details below).

3) Clarity of figures and legends.

Some symbols and colors used in the figures are not well explained in the legends making them difficult to follow. The authors need to define all symbols and colors used as well as explaining several details of how certain analyses were done.

Specific comments:

Abstract: several statements here seemed to me to be overstatements:

line 17: “neural processing of a navigation goal requires the continuous comparison between the current heading and the intended goal direction”— what about cases where a visual target is continuously available. this has been shown to not require such a comparison.

line 19: “completely unexplored” not true. e.g. Webb computational work, Hulse et al., Matheson...Nagel, recent preprints from Wilson and Maimon labs, plus prior work from Maimon lab manipulating the HD signal.

line 25: “we here discovered invertebrate goal-direction neurons”. I think it would be helpful to say what was discovered about them. I thin the field has predicted the existence of gGD neurons for many years.

Introduction: similarly:

line 42 “must compare” not necessarily true when the goal cue is continuously available.

45: GD neurons only described in the mammalian brain: I don't think this is fair because of the citations listed above

47: “the neurons specifically represent the animal's navigation goal by...” would also apply to steering neurons

48: “how spatial goals are represented neuronally is completely unknown”. not true, many predicted models for this and some evidence for them in recent papers.

line 54: “migratory behavior is controlled by a region termed the central complex”. has this been shown experimentally via lesion?

line 58: “these cell have not been described”. several recent studies have shown experimental evidence for GD neurons.

Figure 1G-J: can the correlation between tuning curves be shown as a distribution on a cell-by-cell basis rather than clustering as in I? In H, why do the tuning curves in the top right not shift by 180° if these are HD cells? Are all neurons best described as HD or GD or is there a distribution and can you show this?

1D: what does the purple line represent?

1E: please spell out what is being plotted

line 98-99: what was the criterion for “modified the direction of their angular tuning”. Please clarify if you saw a bimodal distribution or have dividing a smooth distribution into two categories.

line 104: please specify how neurons were placed into one category or the other. S6C looks like an arbitrary split in a smooth distribution.

line 108: could neurons that do not rotate with the compass perturbation encode some other allocentric variable than goal?

line 120: I think it would be helpful to specify in the text here what range of angles was punished.

line 125: “control experiments” please specify what these were and what they showed.

Figure 2F: The time course data is really intriguing, can you show population data for the dynamics of activity following conditioning?

2B: is this one animal or many animals?

line 133: “pdfs yoked to butterfly’s flight behavior” please explain what is meant here.

Fig 3B: what are the activity patterns of the remaining 60% of neurons?

Discussion line 205-206: I think you could say a bit more about anatomy and its relationship to neuronal coding properties. Is the mixture of HD and GD neurons in your sample surprising if you think most of these recordings were from the FB? Was there any relationship between electrode position and response type?

Fig S1: Did you observe any differences in neural encoding depending on whether the animal had a preferred heading closer to the sun direction or further away?

Fig S2: please explain the gray and purple bars in the top plot.

Reviewer #2 (Remarks to the Author):

In this very interesting manuscript, Beetz et al. provide the first evidence for goal-direction coding in the insect brain – a very exciting advance. By recording single neurons in the central complex of navigating (tethered) monarch butterflies, they found both head-direction cells (HD cells), which rotated when the virtual sun was rotated; and a roughly equal number of neurons which maintained a stable goal-direction irrespective of the sun direction. These neurons were defined as goal direction cells (GD cells). These results provide a very important missing piece in the puzzle of navigation neurons in insects – and these findings also shed a very interesting comparative light on the navigation system across taxa, from insects to mammals. These findings are of great importance for the neurobiology of navigation, and for

neuroscience in general, as they resolve a long-standing missing piece in the navigation circuit in the insect brain: these goal-direction cells may be the mechanism by which the animal navigates towards goals, and thus they might play an important role in the amazing navigational feats of monarch butterflies, and other insects. I believe that this paper would be of great interest for neuroscientists in general, whether they study insects or mammals, or other invertebrates or vertebrates. In short, this paper is of broad general interest to the entire systems-neuroscience community. I therefore enthusiastically recommend publication. I have only a few (semi-) major comments and a few minor comments.

Major comments:

1. The authors write that the butterflies were not in the migratory phase (lines 76-77). This raises the question: what makes their goal a “goal”? After all, they are not really intending to migrate towards any goal, if they are not in their migratory phase. So, in which sense is this a goal? Please discuss this issue in the Discussion section.
2. Fig. 1E, x-axis was confusing: what is delta-heading / delta-sun ? Please elaborate here in the figure legend what are these values, and how are they computed. Also, isn't this ratio noisy when delta-sun is small? I suggest that, due to this putative noise problem, the authors should plot, in addition to the current histogram, also another histogram: delta-heading minus delta-sun (difference in degrees).
3. In Fig. 1, I suggest that the authors should add also histograms of preferred heading directions for the compass neurons (HD cells), as well as for the GD neurons. These histograms are important, and should be in the main-text figures.
4. I did not see any shuffling of spike trains as typically done for defining HD cells in mammals (and also for place cells, grid cells, etc.). This is a standard statistical procedure in the literature of mammalian neurobiology of navigation, and should be done here as well. The authors should shift the spike trains randomly and circularly e.g. 1,000 times relative to the true behavior, then for each such shuffle they should compute the mean vector length (MVL), and compare to the MVL of the real directional tuning curve. How many of the neurons had real MVL > 95% of the shuffles?
5. Please add here a histogram of MVL's (mean vector lengths) for the directional tunings of HD cells and GD cells. How strongly tuned are the neurons in Fig. 1H ?

6. Lines 383-384: I didn't understand fully why $HI > 0$ means HD cell and $HI < 0$ means GD cell. Please add an illustration of this concept in a supplementary figure, and explain in great detail the logic of these definitions in the figure legend of that new supplementary figure. I think this is crucial to explain this in detail.

Minor comments:

1. Line 44, in the citation of HD cells across species (refs. 3-12), it would be good to cite also HD cells in monkeys (Angelaki lab: reviewed in PMID 31877492) and in bats (Finkelstein et al. 2105: PMID 25470055).

2. Fig. 1H, color code says "Norm. FR": how is the normalization done? Did the authors divide each tuning curve by its maximum? Or some other normalization? This should be specified in the figure legend.

3. Figure legend of Fig. 1C, a dot is missing at the end of the sentence. This was confusing.

4. Line 92, the authors write that "the tetrodes were mainly positioned in the fan shaped body of the central complex": does this mean that some neurons were recorded from other areas of the central complex? If so, are there any noteworthy differences between the different brain areas? Specifically, any interesting differences in the tuning properties of neurons, proportions of GD cells versus HD cells, sensitivity to perturbations, etc. ?

5. Line 98, "In total, 55 of 113 neurons... modified..." – this was very confusing at this stage of the main text to understand where did this number 55 come from, and how these neurons were defined. Please define in detail, as early as possible in the main text, how did you define the 55 neurons that modified their firing (what does it mean "modified"), and the 58 cells that were unaffected (what does it mean "unaffected"). I know this is explained later, but this explanation must come early. And on the same note: I also think that a more detailed explanation should be given to the dichotomous separation of "modified" neurons – i.e. HD neurons (line 98), versus "unaffected" neurons (line 103) – i.e. GD neurons. This is a critical separation, and should be explained in detail, and early.

6. Line 382: the "HI formula" was confusing initially, because in the main text the authors called this "HD index" (lines 111-112); please call it "HD index" also here, instead of (or in addition to) "HI", in order to prevent confusion.

7. Fig. S1 legend, you wrote twice “circularplot”: this should be “circular plot”, in two words.

Reviewer #3 (Remarks to the Author):

The manuscript reports an exciting and timely set of experiments to examine goal direction coding in the monarch butterfly brain. There is much to commend the authors for in this study. The butterflies studied are famous for their navigation skills and thus recording from them in this manner will be of interest to many readers. The technical challenges to conduct the research are high and the analyses chosen to examine the data are generally rigorous and provide a good degree of examination of the data. I have a number of points I think the authors should address to strengthen the manuscript.

1. From the data presented, the sampling of different directions in the flights is unclear. A core approach in from neural recordings in rodents is to characterise the directional tuning compared to the distribution of possible heading orientations recorded. Rodent/bat head direction (HD) cells show clear tuning to certain directions despite the sampling directions, and in some cases being biased. Typically, that research tries to probe directions of movement evenly dispersed over 360 degrees. From the 3 samples plots of the trajectories, the current data is very biased, due to the natural flying behaviour examined. It will be important for the authors to present much more detailed analyses of the flying trajectories, and their relation to the sun stimulus over the period. Some plots do show the orientations alongside spiking (eg.. Fig S8), but the radial spiking plots do not and this is critical to the interpretation. For the data to be meaningful it is important that the GDs show spiking patterns that are not simply driven by facing most often in a certain direction.

2. Related to point 1, Fig S13 shows the distribution of pfd of HD neurons relative to the butterflies' GDs. This is helpful, but it would also be important to present the HDs and GDs relative to the sun stimulus and relative to the distribution of headings experienced during recording.

3. The authors use 'allocentric heading' in Fig 1. But is this not just heading relative to the sun stimulus? If so, this conflates a light source in an egocentric and an allocentric reference frame. The authors may wish to add extra evidence for considering the data in these terms or change to phrasing.

4. In addition to the literature cited, I suggest the authors also cite studies showing goal approach/direction-selectivity coding:

Gauthier, J. L., & Tank, D. W. (2018). A dedicated population for reward coding in the hippocampus. *Neuron*, 99(1), 179-193.

Aoki, Y., Igata, H., Ikegaya, Y., & Sasaki, T. (2019). The integration of goal-directed signals onto spatial maps of hippocampal place cells. *Cell reports*, 27(5), 1516-1527

And also fMRI evidence for allocentric and egocentric goal direction representations in the human brain that are distinct from facing orientations:

Chadwick, M. J., Jolly, A. E., Amos, D. P., Hassabis, D., & Spiers, H. J. (2015). A goal direction signal in the human entorhinal/subicular region. *Current Biology*, 25(1), 87-92.

Shine, J. P., Valdes-Herrera, J. P., Tempelmann, C., & Wolbers, T. (2019). Evidence for allocentric boundary and goal direction information in the human entorhinal cortex and subiculum. *Nature communications*, 10(1), 1-10.

Reviewer #4 (Remarks to the Author):

The authors need to be careful on how they use terms related to “navigation” and “spatial”. For example, the authors should make it clear that they are studying animal orientation, rather than animal navigation, as navigation implies an individual has the capacity to know its current place relative to a specific destination (e.g., map sense). Orientation and navigation are related, yet distinct concepts. Moreover, if the authors are studying the coding of a “goal direction”, then this is consistent with orientation not navigation, since a “goal direction” can also be simply called a “preferred direction” or “preferred orientation heading”, and is not a specific destination. The so-called “spatial goal” is not an actual, defined place (an actual goal, such as the overwintering sites of migratory monarch butterflies), as directional behavior to this “goal” and the “goal” itself, can be manipulated as seen in this study. As such, it is more appropriate for the authors to describe their work as further examining how animals orient, as they are examining how animals encode a preferred goal heading and how that heading is related to the animal’s actual current heading during movement.

A major goal of the authors in this manuscript is to relate their results to how migratory monarch butterflies might encode goal directionality (see Introduction and Discussion). Unfortunately, as stated by the authors themselves in the Results (lines 76-77), they are not studying migratory animals at all in their work, and therefore it remains unclear how these results actually relate to the orientation mechanisms of migratory butterflies and migratory animals in general. A large body of work has clearly shown that migratory and non-migratory monarch butterflies are fundamentally different from each other, such as at the morphological, behavioral, physiological, and genetic levels. Moreover, recent work (e.g., Tenger-Trolander et al. 2019) has shown that monarchs from commercial stocks and then reared under controlled conditions (as done by the researchers in this current study; lines 230-234) do not orient in the same way when tested in flight experiments, as compared to migratory butterflies caught

in the wild. Therefore, these results appear to be preliminary and have limited applicability to migratory butterflies, especially within the context of orientation during migration. The authors need to study actual migratory monarch butterflies in their experiments, whose migratory directionality is confirmed when caught in the field, in order to see if these neural mechanisms are applicable to these animals during behavior.

Do the animals actually change their heading in relation to the displacement (e.g., 180° sun displacement; Figure 1D) or is this view simply an artifact of the experimenter's concept of space? It would appear that the animals are not changing their heading at all (especially within the confines of the uniform experimental arena), but are actually maintaining their heading relative to their perception of the virtual sun. For example, in Figure 1D, the butterflies are keeping the position of the virtual sun to the left (left of the mean heading as denoted by the purple line). After the experimenter moved the position of the virtual sun, the butterflies then again simply kept the sun in the same position relative to them, i.e., the virtual sun to the left. Effectively, the butterflies did not change their heading relative to the virtual sun, but maintained their subjective heading. Rather than a preferred heading, which suggests a specific direction such as southwards for a migratory monarch butterfly, could this behavior, and its shift in response to moving the virtual sun, reflect a form of positive phototaxis in which monarchs fly towards light regardless of where the light source is, but with some slight skew? It seems that results from Figure 2 support this, as the aversive conditioning (electrical shocks to the butterflies) makes the butterflies fly almost directly away from the virtual sun, who then are displaying negative phototaxis in trials.

We thank all reviewers for the constructive comments that have clearly helped to improve our manuscript. Please, find below our response to the reviewers' comments. The reviewers' points are written in black while our responses are written in green. To facilitate the review process, we copied all major changes related to the reviewer's comments directly below our responses. 'lines' are referred to the clean version of the draft.

Reviewer #1:

Overall the results are exciting and speak to an emerging understanding on the field of how navigational goals are encoded and read out to meet the behavioral needs of diverse animal species. My main concern is that the division in 3 neuronal classes may be too simplified; the authors might do a better job of illustrating the range of cell types and responses in their data set. As well, the introduction and discussion over-emphasize the novelty of the work at the expense of setting the findings appropriately in the larger context of understanding navigational systems. Finally, some aspects of the figures are not well explained in the legends. Overall I think this is a valuable contribution to the field that can be improved with some additional analysis and editing of the text. We thank the reviewer for her/his positive feedback. Please, find below our responses/ improvements to all points.

Major comments:

1) Division of recorded cells into HD and GD neurons.

The manuscript is framed around the discovery and division of CX neurons into heading direction (compass) and goal direction neurons. Figure 1G shows examples and H-J quantify differences in these populations. However, how cells were classified into these two types is not totally obvious. I would expect the heading direction neurons to shift their tuning curves by 180° with the compass perturbation shown in H, but that does not seem to be what happens from the heat-maps in H (top right plot). For the quantifications in I and J I can't tell how seriously to take these comparisons because I am not sure what the criteria were for dividing cells into the two categories. Figure S6C suggests that what look like a smooth gaussian distribution of properties has been divided in half to form two categories of neurons. Overall I think the authors need to do a better job of (1) illustrating the full diversity of neuronal response types, (2) defining explicit criteria by which they have categorized neurons as HD or GD, and (3) explaining their thinking about the diversity of cell and response types and how they might function in navigation. Recent results from other insect species I think suggest that there are likely many different cell types in the CX with a diversity of response properties perhaps play different roles in different behaviors. I think a better statistical description of the distribution of responses seen in the butterfly would contribute to our understanding of this diversity and its potential roles in butterfly navigation behavior.

We thank the reviewer for raising this important point. Before submitting the first draft, we already tried several analyses to distinguish between neurons that "remap" and neurons that exhibit an invariant tuning during compass perturbation. However, this is unfortunately not trivial due to the way we had to design our experiment (we will explain this in more detail below). Because of the interindividual response of butterflies to sun displacements, we did not only turn the virtual sun by 180°, but also by different angular sizes, which was not clearly explained in our previous version. As we included all sun displacements in our analysis, calculating the HD index was the best way to categorize the neurons. The best assessment for this can be seen in the raw data presented in Fig11 and S7B of the new draft's version. Nevertheless, we agree with the reviewer that we did a poor job in explaining the categorization of the neurons and the HD index in our first draft. In our revised version, we therefore thoroughly explain the way of how we categorized the neurons.

On lines 137-145, we state:

'Based on the influence of compass perturbations on the neural tuning, we classified the neurons into two types: i) compass neurons whose angular tuning was linked to the butterflies' heading direction,

such as HD and steering neurons, and ii) putative GD neurons whose angular tuning was not affected by compass perturbations. The classification was quantified by calculating an HD index for each neuron (for details, see Methods, Fig. S6). Positive HD indices were expected for compass neurons that change their angular tuning, represented by the preferred firing direction (pfd), in accordance with the butterfly's change in mean heading (green neurons in Fig. 1H and Fig. S7A, Fig. S6A). In contrast, putative GD neurons should show negative HD indices as their angular tunings were expected to be unaffected by compass perturbations (blue neurons in Fig. 1H and Fig. S7A; Fig. S6B).'

and in the methods, we added on lines 504-522

'Functional classification of neurons (HD-index)

Neural data were considered from three periods, in which the animals showed the highest flight directedness (r). Neurons were categorized regarding their changes in pfd in response to sun displacements. To categorize if the pfd of neurons changed with the animal's heading, we calculated an HD index. First, we calculated the heading offset, which represents the angular difference between pfd and behavioral heading directions. We then computed the circular variance of these heading offsets (CVH) for each neuron. As neurons linked to the animal's heading should change in accordance with the behavior, we expected that their CVH should be relatively low (Fig. S6). In contrast, neurons that are not linked to the animal's heading, such as GD neurons, should reveal relatively high CVH values. In addition, we computed the circular variance of all pfd (CV). The CV should be relatively low for neurons with an invariant tuning, such as GD neurons and high for neurons with a variant tuning, such as compass neurons (Fig. S6). The HD index for each neuron was then calculated by combining the CVH and CV with the following equation:

$$HD\ index = \frac{(CVH - CV)}{(CVH + CV)}$$

An HD index > 0 indicates that the neural tuning can better be explained by a correlation with the animal's heading (putative HD & steering neurons, n = 55), while an HD index < 0 indicates that neural tuning was unaffected by the animal's heading (putative GD neurons, n = 58). In addition, we correlated the binned neural response (10° bin size) measured prior to sun displacement with the one measured after displacement.'

For clarity, we also added a new supplemental figure (Figure S6) that illustrates the calculation of the HD index.

We hope this makes our analysis clear.

Another excellent point raised by the reviewer (and that we did not clearly explain in our previous version) is the lack of 180° shifts in the angular tuning of compass neurons shown in Fig 1I (new version). The reviewer is fully correct: we expect 180° shifts if we always turn the green sun stimulus by 180°. However, we had to turn the virtual sun by different angular sizes, because we knew from previous work that butterflies do not always turn when the virtual sun is displaced by 180° (Franzke et al. 2022, JEB), most likely because we create a conflict between idiothetic and allothetic information during compass perturbations (Beetz et al. 2022, Curr Biol). As this conflict is lower when we turn the virtual sun by smaller angular sizes, different angular sizes of sun displacements were necessary to reliably change the butterfly's mean heading. We now clearly state which angular sizes were used to displace the virtual sun on lines 89-90: *"This was achieved by displacing the virtual sun along the azimuth at different angular sizes, i.e., 180°, 90°, 45°, 25°, and 15°, every 90 s (Figs. 1C, S2)."*

As not all butterflies responded equally to 180° turns of the virtual sun, as explained above, plotting only 180° sun displacements would not have been very meaningful. We therefore considered only three periods of the compass perturbation in which an individual butterfly showed the highest flight

directedness, i.e., the longest flight vector. This ensures that we analyzed only periods in which the butterflies were motivated to maintain a goal direction relative to the virtual sun. Comparable filters had also been set in Pires et al. (2022, Biorxiv) to study menotactic behavior in fruit flies. Two of these three periods are plotted in Fig. 1I of the revised version. As these periods represent angular tunings in response to different angular sizes of sun displacements, e.g., 15°, 25°, 45°, 90° or 180°, a 180° shift in angular tuning was not expected. This information is now added to the figure legend on lines 121-122:

“Note that the neural tuning of the most directed flight sections of the butterfly (i.e., either 15°, 25°, 45°, 90° or 180° sun displacement) to the sun displacement is plotted”.

However, one major problem of the visualization in Fig1I is that it does not show whether the compass neurons’ angular tuning is linked to the animals’ mean heading. To visualize this, we plotted the angular tunings of the neurons shown in Fig 1I, in relation to the animal’s mean heading direction (Fig. S7B). As evident from this figure, the tuning of the compass neurons is linked to the animal’s heading direction. We added the following information in the results on lines 150-152: *‘The strong link between the animal’s heading and spatial tuning of compass neurons is apparent when the neurons’ firing rate is plotted relative to the butterflies’ mean heading (0° represents the animal’s heading direction; Fig. S7B)’*

Regarding a more detailed neuronal categorization: Our experiments, unfortunately, do not allow us to functionally discern each neuron type of the central complex, simply because the experiments were not designed for this. We designed our experiments with the aim to describe how the insect goal direction is encoded. Consequently, it allows us to distinguish between neurons encoding the insect’s heading direction (HD neurons), steering direction (possibly PFL neurons), and goal direction (GD neurons). In addition to these neurons, we also found neurons that did not change their coding during both compass perturbation and aversive conditioning (please, see in the methods on lines 556-558). These neurons could be cells that encode a cue in an absolute frame of reference, such as a magnetic cue. However, as we did not control for any magnetic field in our experiments this is highly speculative, and we therefore stated them only in the Methods.

While a classification into three types of neurons may oversimplify the number of different types of central-complex neurons and their functions, our experiments do not allow us to reliably make a more detailed separation. Many central-complex neurons, such as E-PG, delta7 and P-EN neurons are expected to change their tuning due to compass perturbations but should not show a change when resetting the goal direction. Moreover, we cannot exclude that the GD neurons may represent multiple anatomically different cell types. Here, recent, and future results in *Drosophila* (see Pires et al. 2022 biorxiv; and Matheson et al. 2021 Nat comm.) may provide us with more neural parameters to functionally discern different classes of central complex neurons in non-model organisms.

2) Framing of the manuscript within the context of recent literature.

The authors would like to draw attention to the novelty of their finding of goal direction neurons, but I think in doing so they mis-represent the current state of the literature. For example, the abstract states that the coding of goal direction is “completely unexplored insects” which is not true. Multiple computational studies have proposed the existence and nature of goal signals, and experimental evidence for goal signals in insects has been shown recently through optogenetic activation in flies as well as recordings in more than one study. Many novelty statements in the Abstract, first paragraph of the Introduction, and Discussion should be toned down and set in a more appropriate context of other studies (see details below).

It was not our intention to ignore any data on goal coding in insects, especially not the computational studies that have clearly served as a crucial basis to guide our study. As the reviewer her/himself states,

the existence of GD neurons has been predicted for many years and we wanted to emphasize that we now provide *empirical* data on the coding of goal direction in the insect brain. The two *Drosophila* preprints [Pires et al. (2022) and Westeinde et al. (2023)] have been uploaded on *BioRxiv* after our initial submission to Nat Comm. Therefore, we were not able to discuss our findings in the light of these papers (which we are doing now – for more information, see minor points).

Matheson et al. (2021) did not manipulate the insect's goal direction and imaged from hΔc cells. Therefore, it remains unclear to what extent these cells control the insect's goal direction - see also the recent review by the same lab [Steele et al. (2023), *J Comp Physiol A*].

We carefully went through our manuscript and made sure that we do not mis-represent the current state of literature and have incorporated all papers suggested by the reviewer. We thank the reviewer for pointing this out and hope that she/he is satisfied with how we now discuss our results in the new manuscript version. For more information, please, see minor comments.

3) Clarity of figures and legends.

Some symbols and colors used in the figures are not well explained in the legends making them difficult to follow. The authors need to define all symbols and colors used as well as explaining several details of how certain analyses were done.

We carefully went through all figure legends and added more details on the color-codes and used symbols. In general, we added more information to the results to make them more comprehensive.

Specific comments:

Abstract: several statements here seemed to me to be overstatements:

line 17: “neural processing of a navigation goal requires the continuous comparison between the current heading and the intended goal direction”— what about cases where a visual target is continuously available. this has been shown to not require such a comparison.

This point is also related to a point raised by another reviewer. We here studied how a desired heading direction is encoded, rather than a specific goal location. To avoid any misinterpretation of our experiments, we have rephrased this sentence (lines 17-18):

“Neural processing of a desired moving direction requires the continuous comparison between the current heading and the goal direction.”

line 19: “completely unexplored” not true. e.g. Webb computational work, Hulse et al., Matheson...Nagel, recent preprints from Wilson and Maimon labs, plus prior work from Maimon lab manipulating the HD signal.

The reviewer is correct. We rephrased the sentence.

However, to the best of our knowledge our draft and the recent preprint by the Maimon lab are the only studies that recorded/imaged GD neurons in insects. It is unclear whether the neurons by Matheson et al. specifically control the insect's goal direction or whether they generally trigger goal directed olfactory navigation. However, we included the following information in the introduction and the discussion to put our findings into a broader context. Similarly, the data from the Wilson lab were mostly focusing on neurons postsynaptic to GD neurons such as PFL cells, which – although they also carry information about goal directions – cannot be described as GD neurons. Nevertheless, we have incorporated all papers in our manuscript:

Lines 59-65: *‘To be able to control for steering, the central complex needs additional information on the insect's goal direction as proposed in anatomical and computational studies^{22,23,25-27,42,44,45}. Because manipulations of the neural activity in HD neurons did not affect the goal direction in fruit flies³, goal directions must be processed by different central-complex neurons. Notably, a recent study shows that*

*the activity of a set of central-complex neurons triggers a goal-directed orientation in fruit flies*⁴². However, to what extent the activity of these neurons dictates the fly's goal direction remained unclear^{42,46}.'

In the discussion we added on lines 291-296: *'Recent studies predict that GD neurons innervate the fan-shaped body of the central complex*^{25,26,42,45,58} *which fits well with our recording site (Fig. S4). However, as most central-complex neurons, including HD and steering neurons, also project through the fan-shaped body*⁸⁰, *our technique does not allow us to conclude where exactly these neurons are localized in the central complex. Interestingly, a recent study confirms the existence of GD neurons, termed FC2 neurons, in the Drosophila fan-shaped body*⁵⁸. *If our GD neurons are similar cells as the FC2 neurons, remains to be determined in the future.'*

In the discussion, we also added the preprint by the Wilson lab (lines 322-330): *'The representation of the current heading in the ellipsoid body and the goal direction in the fan-shaped body are thought to be compared by downstream steering neurons*^{25,26,45}. *Recent findings in Drosophila suggest that this comparison is carried out by two different neuron types, termed PFL3 and PFL2*^{42,43,58}. *PFL3 neurons are active and induce steering commands whenever the insect deviates from its intended goal direction, whereas PFL2 neurons are primarily active when the insect heads into the anti-goal direction*⁴³. *This is well in line with the monarch butterfly steering neurons described here and suggests that these neurons transfer motor commands to descending neurons whenever the butterfly is flying into the anti-goal direction.'*

Please acknowledge that both preprints were submitted after our initial submission and, hence, we had no chance to embed them in our first draft.

line 25: "we here discovered invertebrate goal-direction neurons". I think it would be helpful to say what was discovered about them. I think the field has predicted the existence of GD neurons for many years.

Due to space limit (abstracts in *Nat Comm* cannot exceed 150 words), we have added the requested information to the introduction on lines 71-76: *'Remarkably, another subset of neurons was specifically tuned to the butterflies' goal-direction: When we conditioned the butterflies to set a new goal direction, the angular tuning of these neurons accurately followed the behavioral change in goal direction implying that these neurons represent GD neurons. Moreover, by simultaneously recording from GD and steering neurons, we found that the GD neurons are suited to initiate turns back to the goal heading whenever the animal was facing away from its goal direction.'*

Introduction: similarly:

line 42 "must compare" not necessarily true when the goal cue is continuously available.

We rephrased this sentence.

45: GD neurons only described in the mammalian brain: I don't think this is fair because of the citations listed above.

We now specify the statement as follows (lines 45-48) *'[...], goal-direction (GD) neurons have only been empirically recorded in the mammalian brain*¹⁵⁻²¹ *[...] Although, GD neurons had been predicted in several theoretical studies*²²⁻²⁷ *it is still not fully clear, how the goal direction is encoded in insects.'*

While computational studies have predicted GD neurons, their physiology was not known so far. The data in Pires et al. (2022) are highly consistent with our findings and have therefore been stated in the discussion (lines 270-273). *'This specific coding of the goal direction is well in line with very recent findings from the Drosophila brain*⁵⁸ *and confirms that the insect brain houses GD neurons similar to the ones previously discovered in the mammalian hippocampus*¹⁵'

As stated above, this paper was uploaded on BioRxiv while our paper was under review, and it is still not published in a peer-reviewed journal. We therefore believe that the most appropriate way is to cite it in the discussion.

47: “the neurons specifically represent the animal’s navigation goal by...” would also apply to steering neurons.

This sentence refers to the findings in bats, where GD neurons specifically represent the bat’s direction to the goal. In insects, it is not straightforward to functionally disentangle between GD and steering neurons. Therefore, it was necessary to perturb the insect’s compass without changing the insect’s goal direction, as we did in the present study. In this case, steering neurons changed their angular tuning (because the steering direction changed) BUT the angular tuning of GD neurons was unaffected. To avoid any confusion at the beginning of the introduction, we decided to not explicitly explain the expected differences in goal and steering coding.

48: “how spatial goals are represented neuronally is completely unknown”. not true, many predicted models for this and some evidence for them in recent papers.

We have made substantial changes to the paper regarding this and added several papers to our manuscript (see previous points).

line 54: “migratory behavior is controlled by a region termed the central complex”. has this been shown experimentally via lesion?

We rephrased this sentence (lines 53-55): *‘Given the highly conserved function of the insect central complex^{1,35-37}, this brain region is most likely central for spatial orientation in monarch butterflies.’*

line 58: “these cells have not been described”. several recent studies have shown experimental evidence for GD neurons.

We have rephrased this sentence.

Figure 1G-J: can the correlation between tuning curves be shown as a distribution on a cell-by-cell basis rather than clustering as in I? In H, why do the tuning curves in the top right not shift by 180° if these are HD cells? Are all neurons best described as HD or GD or is there a distribution and can you show this?

We thank the reviewer for pointing this out: We changed the subfigure accordingly and replotted the data in reference to the animal’s body axis. This demonstrates that angular tuning of compass neurons could best be explained by referring to the compass direction (Fig. S7B). We are also plotting now the remaining correlations (Fig. 3D and 3E) in a cell-by-cell manner in the revised version. Regarding the 180° shift, please see our responses to the reviewer’s major comments.

1D: what does the purple line represent?

As depicted in the figure it shows the insect’s mean heading. We added the info also in the figure legend.

1E: please spell out what is being plotted

Done.

line 98-99: what was the criterion for “modified the direction of their angular tuning”. Please clarify if you saw a bimodal distribution or have dividing a smooth distribution into two categories.

We now thoroughly explain our categorization in the revised version (see major point 1).

line 104: please specify how neurons were placed into one category or the other. S6C looks like an arbitrary split in a smooth distribution.

We fully agree with the reviewer's concerns and carefully assessed our neuronal categorization by plotting the raw data from all neurons (see heatmaps in Fig. 1 and Fig. 3). Most importantly, the categorization aimed to extract GD-neurons that showed an invariant tuning during compass perturbation and predictably changed their tuning in accordance with the insect's change in goal direction (Fig. 3B upper row of heatmaps). When considering the heatmaps in figure 3B and the HD indices calculated from the angular tuning during aversive conditioning (Fig. S6D), we believe that it becomes clear that our classification via the HD index is a robust way to distinguish GD neurons from other cells.

line 108: could neurons that do not rotate with the compass perturbation encode some other allocentric variable than goal?

As mentioned in the draft, another plausible explanation could be the coding of geomagnetic information. Therefore, we emphasized on the importance of combining both compass perturbations and aversive conditioning. Only in this way, we can explicitly state that we recorded from GD neurons.

line 120: I think it would be helpful to specify in the text here what range of angles was punished.

We added the following information (lines 172-173): *'[...] by applying small electric shocks to the butterflies' necks whenever they headed towards their initial goal direction ($\pm 90^\circ$; Fig. 2A).'*

line 125: "control experiments" please specify what these were and what they showed.

We are now explaining the control experiments in more detail in the revised version (lines 185-188): *'We further excluded an effect of electric stimulation on the angular tuning through control experiments in which we showed that electric stimulation of central-complex neurons in restrained butterflies does not change the angular tuning ($p = 0.63$, $W = 1136$, $n = 256$, Wilcoxon matched-pairs signed rank test, Fig. S11).'*

Figure 2F: The time course data is really intriguing, can you show population data for the dynamics of activity following conditioning?

Thank you for appreciating the time course data. We would love to do a more thorough analysis on the temporal dynamics of angular tuning. Unfortunately, the sliding window for computing the time course data was extremely large (several minutes). This was necessary because the butterfly had to steer in each direction to calculate a pfd (which is not trivial if they maintain a certain goal direction). Using such a large time window substantially degrades the temporal precision represented in the data. Therefore, we can use the time course data only for visualization purposes and decided to present only two examples in the draft (Fig. 2G, S13A).

2B: is this one animal or many animals?

This is just one representative butterfly. We added this information in the figure legends for figures 2 and 1.

line 133: "pdfs yoked to butterfly's flight behavior" please explain what is meant here.

We replaced the term 'yoked' by 'linked to'.

Fig 3B: what are the activity patterns of the remaining 60% of neurons?

Altogether, we recorded from 65 neurons showing angular sensitivity throughout compass perturbation and aversive conditioning. From these, 20 neurons were classified as GD neurons, 13 as

HD neurons, 19 as steering neurons (Fig. 4). The remaining 13 neurons showed an invariant tuning during compass perturbation AND conditioning (see methods). Because we could not explicitly answer their function, we decided to not describe them in the results (see point above). We are currently designing a new setup that will allow us to study the coding of magnetic information in the future.

Discussion line 205-206: I think you could say a bit more about anatomy and its relationship to neuronal coding properties. Is the mixture of HD and GD neurons in your sample surprising if you think most of these recordings were from the FB? Was there any relationship between electrode position and response type?

No, there was no relationship between electrode position and response type. In general, we do, on purpose, not like to go into more details regarding the tetrode position. We believe it is not very meaningful as it is still unclear how local our extracellular recordings are. This is also the reason why we measured the angular tuning when the animal oriented in darkness prior to flight. This allows us to study only neurons that internally represent directions as expected from central-complex neurons. Combined with the tetrode stainings, we are confident that we are recording from central-complex neurons. In addition to technological limitations, the neural fibers of most central-complex neurons run through the fan-shaped body as shown in fig. S4. Thus, we could also pick up neurons from the ellipsoid and fan-shaped body with the same tetrode, despite the tetrode being placed in the fan-shaped body (which is most likely the case – otherwise we would have recorded substantially fewer HD neurons in our experiments). In the new version of our manuscript, we state this technological limitation clearly in the discussion (lines 291-294):

'Recent studies predict that GD neurons innervate the fan-shaped body of the central complex^{25,26,42,45,58} which fits well with our recording site (Fig. S4). However, as most central-complex neurons, including HD and steering neurons, also project through the fan-shaped body⁸⁰, our technique does not allow us to conclude where exactly these neurons are localized in the central complex.'

Fig S1: Did you observe any differences in neural encoding depending on whether the animal had a preferred heading closer to the sun direction or further away?

This is a very good question. So far, we could not find any differences in the neural tuning depending on the insect's preferred heading relative to the sun direction.

Fig S2: please explain the gray and purple bars in the top plot.

We explain gray and purple bars in the figure legends (also in Figure 1) in our revised version.

Reviewer #2:

In this very interesting manuscript, Beetz et al. provide the first evidence for goal-direction coding in the insect brain – a very exciting advance. By recording single neurons in the central complex of navigating (tethered) monarch butterflies, they found both head-direction cells (HD cells), which rotated when the virtual sun was rotated; and a roughly equal number of neurons which maintained a stable goal-direction irrespective of the sun direction. These neurons were defined as goal direction cells (GD cells). These results provide a very important missing piece in the puzzle of navigation neurons in insects – and these findings also shed a very interesting comparative light on the navigation system across taxa, from insects to mammals. These findings are of great importance for the neurobiology of navigation, and for neuroscience in general, as they resolve a long-standing missing piece in the navigation circuit in the insect brain: these goal-direction cells may be the mechanism by which the animal navigates towards goals, and thus they might play an important role in the amazing navigational feats of monarch butterflies, and other insects. I believe that this paper would be of great interest for neuroscientists in general, whether they study insects or mammals, or other invertebrates or vertebrates. In short, this paper is of broad general interest to the entire systems-neuroscience

community. I therefore enthusiastically recommend publication. I have only a few (semi-) major comments and a few minor comments.

We are grateful for the reviewer's very positive feedback.

Major comments:

1. The authors write that the butterflies were not in the migratory phase (lines 76-77). This raises the question: what makes their goal a "goal"? After all, they are not really intending to migrate towards any goal, if they are not in their migratory phase. So, in which sense is this a goal? Please discuss this issue in the Discussion section.

We agree with the reviewer that, the way the previous version of the manuscript was written was slightly misleading, as it sounded as if we studied the navigation of the non-migratory butterflies towards a specific goal. However, this was not studied here. Many insects maintain an individual-specific goal direction relative to a compass cue, which is termed menotactic orientation. These animals therefore keep a constant direction without aiming for a specific location which likely supports optimal dispersal. To clarify this, we went through the entire manuscript and rephrased the information where applicable. In addition, we added the following information on lines 82-87:

'Although the tested monarch butterflies were non-migratory, they reliably maintained goal directions (Fig. 1B), consistent with recent behavioral findings⁴⁷. Across individuals, the goal directions were arbitrary (Fig. S1) resembling menotactic orientation behavior observed in a variety of insects including dung beetles⁴⁸ and fruit flies^{39,49}. Menotactic orientation optimizes animal dispersal⁴⁸ and can be expected to emerge by matching the current heading – encoded by the butterflies' compass – with an internal goal representation^{2,3}.'

2. Fig. 1E, x-axis was confusing: what is delta-heading / delta-sun? Please elaborate here in the figure legend what are these values, and how are they computed. Also, isn't this ratio noisy when delta-sun is small? I suggest that, due to this putative noise problem, the authors should plot, in addition to the current histogram, also another histogram: delta-heading minus delta-sun (difference in degrees).

We spelled out the x-axis title (Fig. 1E in revised version) and explained the calculation of the ratio in the main text. Moreover, we also calculated the difference in degrees (Fig. S3F) which are consistent with the findings of calculating the ratios. We added in the results on lines 98-99:

'Consistently, the angular size of sun displacements subtracted from the change in heading direction was clustered around 0° (Fig. S3F).'

3. In Fig. 1, I suggest that the authors should add also histograms of preferred heading directions for the compass neurons (HD cells), as well as for the GD neurons. These histograms are important, and should be in the main-text figures.

Great suggestion. The pdfs of HD and GD neurons are uniformly distributed ($p = 0.76$ and 0.36 , Rayleigh test) and are now presented in Fig. 1L. The pdfs are plotted relative to the position of the sun (and not relative to the insect's goal direction, as shown in Fig. 4G).

4. I did not see any shuffling of spike trains as typically done for defining HD cells in mammals (and also for place cells, grid cells, etc.). This is a standard statistical procedure in the literature of mammalian neurobiology of navigation, and should be done here as well. The authors should shift the spike trains randomly and circularly e.g. 1,000 times relative to the true behavior, then for each such shuffle they should compute the mean vector length (MVL), and compare to the MVL of the real directional tuning curve. How many of the neurons had real MVL > 95% of the shuffles?

This is an excellent suggestion. In our revised version, we shuffled the spike trains 1,000,000 times per neuron and statistically compared the MVLs of the shuffled data with the MVLs obtained from the real data within neurons (right histograms in Fig. 1H exemplarily shown for the representative example

neurons). Moreover, we compared the MVLs of the shuffled data pairwise with the MVLs of the real data from the entire neural population (Fig. 1G). In both comparisons, the MVLs computed from shuffled data were statistically shorter than the MVLs of the real data ($p < 10^{-5}$, $W = -10878$, $n = 147$, Wilcoxon-signed rank match test). The distribution of MVLs for all neurons showing in angular sensitivity ($n = 113$; $p < 0.05$ Rayleigh test) are plotted in the new figure S5C.

We added the following information in the result section of the revised draft (lines 132-136): *'The tuning directedness to the virtual sun, represented by the mean vector length (MVL; mean \pm standard deviation: 0.15 ± 0.09 ; Fig. S5C), was statistically longer than the tuning directedness modelled from shuffled data (Fig. 1G, $p < 10^{-5}$, $W = -10878$, $n = 147$, Wilcoxon matched-pairs signed rank test, see Methods for details), suggesting that the neurons exhibited a directed angular tuning during flight.'*

We also added the calculation of the MVL in the methods (lines 494-502):

'Tuning directedness of the neurons was quantified by calculating the mean vector length (MVL) of angular tuning (Rayleigh statistics). The MVL of each neuron was statistically compared to a distribution of MVLs generated by a permutation (1, 000, 000 repetitions). The permutation shuffled the 10° bins of the circular plots and computed a MVL after each shuffle. MVL of the real data were significantly longer than the MVLs computed from the shuffled data from each neuron ($p < 0.05$; Fig. 1L). In addition to the neuron-wise comparison of MVL and MVLs from the shuffled data, we statistically compared the measured MVLs from all the neurons with the MVLs after shuffling the data from all 113 neurons ($p < 10^{-5}$, $W = -10878$, $n = 147$, Wilcoxon matched-pairs signed rank test).'

5. Please add here a histogram of MVL's (mean vector lengths) for the directional tunings of HD cells and GD cells. How strongly tuned are the neurons in Fig. 1H ?

We added a histogram as suggested in Fig. S5C of the revised version (plus the result of the MVL for the shown neurons in Fig. 1H). There, we pooled data from HD and GD neurons whose MVLs were statistically not different.

6. Lines 383-384: I didn't understand fully why $HI > 0$ means HD cell and $HI < 0$ means GD cell. Please add an illustration of this concept in a supplementary figure, and explain in great detail the logic of these definitions in the figure legend of that new supplementary figure. I think this is crucial to explain this in detail.

Because we expected that the angular tuning of HD neurons follows the animal's heading after each sun displacement, we expected that their coding changes in accordance with the insect's change in mean heading (HD index > 0). In contrast, we expected that the tuning of GD neurons was invariant irrespective of sun displacements (negative HD index), because the insect's goal direction relative to the virtual sun was consistent. See also the illustration in Fig. S6 together with the figure legend.

In the main text of the revised version, we elaborated the categorization of the neurons and also illustrated it in the new Fig. S6. Its legend says: *'Figure S6. Categorization of neurons based on the HD index. (A, B) Schematic visualization of the HD index calculation used to categorize compass neuron and GD neurons. A hypothetical angular tuning expected from a compass neuron (A) and a GD neuron (B) is shown. The circular variance of heading offsets (CVH) represents the variance of the angular relationship between the butterfly's mean heading (magenta bars) and the neurons' pfd's across the compass perturbations (green for compass neuron and blue for GD neuron). The CVH is expected to be smaller for compass neurons than for GD neurons.'*

As the butterfly's goal direction relative to the virtual sun was consistent in response to compass perturbations, angular tuning of GD neurons was expected to be invariant. The invariance of angular tuning was assessed by calculating the circular variance of pfd's (CV) which was expected to be lower in GD neurons than in compass neurons. The HD index combines the neurons' CVH and CV. Neurons whose angular tuning can best be explained when considering the butterfly's heading direction have

positive HD indices (A), while neurons with an invariant angular tuning may result in negative HD indices. (C, D) Histograms of measured HD indices from neurons recorded during compass perturbation (C, n = 113) and the aversive conditioning (D, n = 65).'

On lines 137-145, we state:

'Based on the influence of compass perturbations on the neural tuning, we classified the neurons into two types: i) compass neurons whose angular tuning was linked to the butterflies' heading direction, such as HD and steering neurons, and ii) putative GD neurons whose angular tuning was not affected by compass perturbations. The classification was quantified by calculating an HD index for each neuron (for details, see Methods, Fig. S6). Positive HD indices were expected for compass neurons that change their angular tuning, represented by the preferred firing direction (pfd), in accordance with the butterfly's change in mean heading (green neurons in Fig. 1H and Fig. S7A, Fig. S6A). In contrast, putative GD neurons should show negative HD indices as their angular tunings were expected to be unaffected by compass perturbations (blue neurons in Fig. 1H and Fig. S7A; Fig. S6B).'

and in the methods, we added on lines 504-522

'Functional classification of neurons (HD-index)

Neural data were considered from three periods, in which the animals showed the highest flight directedness (r). Neurons were categorized regarding their changes in pfd in response to sun displacements. To categorize if the pfd of neurons changed with the animal's heading, we calculated an HD index. First, we calculated the heading offset, which represents the angular difference between pfd and behavioral heading directions. We then computed the circular variance of these heading offsets (CVH) for each neuron. As neurons linked to the animal's heading should change in accordance with the behavior, we expected that their CVH should be relatively low (Fig. S6). In contrast, neurons that are not linked to the animal's heading, such as GD neurons, should reveal relatively high CVH values. In addition, we computed the circular variance of all pfd (CV). The CV should be relatively low for neurons with an invariant tuning, such as GD neurons and high for neurons with a variant tuning, such as compass neurons (Fig. S6). The HD index for each neuron was then calculated by combining the CVH and CV with the following equation:

$$HD\ index = \frac{(CVH - CV)}{(CVH + CV)}$$

An HD index > 0 indicates that the neural tuning can better be explained by a correlation with the animal's heading (putative HD & steering neurons, n = 55), while an HD index < 0 indicates that neural tuning was unaffected by the animal's heading (putative GD neurons, n = 58). In addition, we correlated the binned neural response (10° bin size) measured prior to sun displacement with the one measured after displacement.'

Minor comments:

1. Line 44, in the citation of HD cells across species (refs. 3-12), it would be good to cite also HD cells in monkeys (Angelaki lab: reviewed in PMID 31877492) and in bats (Finkelstein et al. 2105: PMID 25470055).

We added the suggested papers to the manuscript.

2. Fig. 1H, color code says "Norm. FR": how is the normalization done? Did the authors divide each tuning curve by its maximum? Or some other normalization? This should be specified in the figure legend.

We normalized each tuning curve against its peak. We added this information to the corresponding figure legend.

3. Figure legend of Fig. 1C, a dot is missing at the end of the sentence. This was confusing. Thank you for spotting this. We fixed it.

4. Line 92, the authors write that “the tetrodes were mainly positioned in the fan shaped body of the central complex”: does this mean that some neurons were recorded from other areas of the central complex? If so, are there any noteworthy differences between the different brain areas? Specifically, any interesting differences in the tuning properties of neurons, proportions of GD cells versus HD cells, sensitivity to perturbations, etc. ?

Although we appreciate the reviewer’s suggestions, we do not like to focus too much on the tetrode position during recording, as we believe it is not very meaningful. First, it is still unclear how local our extracellular recordings are. Therefore, we measured the angular tuning when the animal oriented in darkness prior to flight which allows us to select only neurons that internally represent directions (in absence of visual input) and that can be found in the insect central complex. Combined with the tetrode stainings, we are confident that we are recording from central-complex neurons. Second, in addition to technological limitations, the neural fibers of most central-complex neurons run through the fan-shaped body as shown in fig. S4. Thus, we could also pick up neurons from the ellipsoid and fan shaped body in the same tetrode, despite the tetrode being placed in the fan-shaped body (which is most likely the case in our experiments – otherwise we would not have recorded any HD neurons). In the new version of our manuscript, we state this technological limitation clearly in the discussion (line 291-294):

‘Recent studies predict that GD neurons innervate the fan-shaped body of the central complex^{25,26,42,45,58} which fits well with our recording site (Fig. S4). However, as most central-complex neurons, including HD and steering neurons, also project through the fan-shaped body⁸⁰, our technique does not allow us to conclude where exactly these neurons are localized in the central complex.’

5. Line 98, “In total, 55 of 113 neurons... modified...” – this was very confusing at this stage of the main text to understand where did this number 55 came from, and how these neurons were defined. Please define in detail, as early as possible in the main text, how did you define the 55 neurons that modified their firing (what does it mean “modified”), and the 58 cells that were unaffected (what does it mean “unaffected”). I know this is explained later, but this explanation must come early. And on the same note: I also think that a more detailed explanation should be given to the dichotomous separation of “modified” neurons – i.e. HD neurons (line 98), versus “unaffected” neurons (line 103) – i.e. GD neurons. This is a critical separation, and should be explained in detail, and early.

In our revised version, we elaborated on the neural classification and the reasoning behind a dichotomous separation. Please see also our response to major comment 6. While our dichotomous separation may seem oversimplified when considering the number of different types of central-complex neurons, the heatmaps presenting the angular tuning of different neuron types demonstrate that they reflect well our expected findings, i.e., GD-neurons only change their pfd when the butterfly’s goal direction is reset (Fig. 3B upper row of heatmaps). However, we cannot exclude the possibility that the GD neurons may represent multiple anatomically different cell types. Here, recent, and future results in *Drosophila* (see Pires et al. 2022 and Matheson et al. 2021) may provide us with more neural parameters to functionally discern different classes of central-complex neurons in non-model organisms in the future.

6. Line 382: the “HI formula” was confusing initially, because in the main text the authors called this “HD index” (lines 111-112); please call it “HD index” also here, instead of (or in addition to) “HI”, in order to prevent confusion.

We are now using HD index throughout the entire manuscript.

7. Fig. S1 legend, you wrote twice “circularplot”: this should be “circular plot”, in two words.
Thank you so much for pointing this out. We changed it.

Reviewer #3:

The manuscript reports an exciting and timely set of experiments to examine goal direction coding in the monarch butterfly brain. There is much to commend the authors for in this study. The butterflies studied are famous for their navigation skills and thus recording from them in this manner will be of interest to many readers. The technical challenges to conduct the research are high and the analyses chosen to examine the data are generally rigorous and provide a good degree of examination of the data. I have a number of points I think the authors should address to strengthen the manuscript.

We thank the reviewer for her/his positive feedback.

1. From the data presented, the sampling of different directions in the flights is unclear. A core approach in from neural recordings in rodents is to characterise the directional tuning compared to the distribution of possible heading orientations recorded. Rodent/bat head direction (HD) cells show clear tuning to certain directions despite the sampling directions, and in some cases being biased. Typically, that research tries to probe directions of movement evenly dispersed over 360 degrees. From the 3 samples plots of the trajectories, the current data is very biased, due to the natural flying behaviour examined. It will be important for the authors to present much more detailed analyses of the flying trajectories, and their relation to the sun stimulus over the period. Some plots do show the orientations alongside spiking (eg.. Fig S8), but the radial spiking plots do not and this is critical to the interpretation. For the data to be meaningful it is important that the GDs show spiking patterns that are not simply driven by facing most often in a certain direction.

We agree that a uniform distribution of behavioral headings may be optimal to characterize directional tuning, however, as correctly stated by the reviewer, the directed flights of the butterflies are by nature biased in particular heading directions. To minimize the risk that this heading bias affects the angular tuning, we quantified the angular tuning by calculating the mean firing rates at different headings, instead of computing circular histograms from spike events. Moreover, only flight trajectories in which all headings (10° bins) were represented during each period of measured angular tuning (during compass perturbation and during conditioning) were considered for calculating the angular tuning of neurons.

In addition to this, a substantial number of neurons did not vary their pfd's although the animal's mean heading changed substantially. For example, the blue neuron in Fig. 1H shows an invariant pfd, although the butterfly substantially changed its heading direction (biased distribution of headings) after compass perturbation (Fig. 1D). Notably, both example neurons in Fig. 1H belong to the behavioral data presented in Fig. 1D. The same is true for the neurons shown in Fig. 2F and G (corresponding behavioral data plotted in Fig. 2E) and Fig. 3F. Because both GD and HD neurons showed an invariant tuning in spite of substantial changes in heading (compass perturbation for GD and conditioning for HD neurons), we are confident that the directional bias of heading did not affect the neural tuning.

To clarify this, we added the following section to the methods (lines 459-466): *‘To characterize the spatial coding in the tethered butterflies, the tested animals should have ideally explored all possible heading directions uniformly. However, when the goal is to maintain a directed course and the animal's heading is thus biased towards the goal direction, as it was the case in the current study, a uniform heading representation is experimentally impossible⁸. To minimize the effect of a biased heading representation on the angular tuning, we quantified the angular tuning based on the neurons' mean firing rates instead of accumulating single spike events. Moreover, during the experiments, the*

butterflies had to explore each possible heading direction to allow us to calculate the spatial tuning. We found no hints that the behavioral bias in heading affected the neural tuning.'

2. Related to point 1, Fig S13 shows the distribution of pfd of HD neurons relative to the butterflies' GDs. This is helpful, but it would also be important to present the HDs and GDs relative to the sun stimulus and relative to the distribution of headings experienced during recording.

We thank the reviewer for this suggestion and plotted the pfd of GD and compass neurons relative to the sun stimulus in Fig. 1L of the revised version. As expected, the pfd of both neuron types tiled the angular space uniformly ($p = 0.76$; $p = 0.36$; Rayleigh test), which stands in contrast to the pfd of GD neurons when plotted relative to the insect's goal direction (Fig. 4G).

3. The authors use 'allocentric heading' in Fig 1. But is this not just heading relative to the sun stimulus? If so, this conflates a light source in an egocentric and an allocentric reference frame. The authors may wish to add extra evidence for considering the data in these terms or change to phrasing.

This is an important point that the reviewer raises here. The circular plots and heatmaps in the main manuscript are visualized in an allocentric frame of reference. To clarify this, we added a small schematic above the heatmaps in the revised version (Fig. 1I). 0° was a fixed reference direction in the setup (allocentric). We deleted the term 'allocentric' from the x-axes in the main figures to avoid confusions. Heatmaps presented in an egocentric frame of reference, i.e., relative to the animal's body axis are now plotted in Fig. S7B of the revised version.

4. In addition to the literature cited, I suggest the authors also cite studies showing goal approach/direction-selectivity coding:

Gauthier, J. L., & Tank, D. W. (2018). A dedicated population for reward coding in the hippocampus. *Neuron*, 99(1), 179-193.

Aoki, Y., Igata, H., Ikegaya, Y., & Sasaki, T. (2019). The integration of goal-directed signals onto spatial maps of hippocampal place cells. *Cell reports*, 27(5), 1516-1527

And also fMRI evidence for allocentric and egocentric goal direction representations in the human brain that are distinct from facing orientations:

Chadwick, M. J., Jolly, A. E., Amos, D. P., Hassabis, D., & Spiers, H. J. (2015). A goal direction signal in the human entorhinal/subicular region. *Current Biology*, 25(1), 87-92.

Shine, J. P., Valdes-Herrera, J. P., Tempelmann, C., & Wolbers, T. (2019). Evidence for allocentric boundary and goal direction information in the human entorhinal cortex and subiculum. *Nature communications*, 10(1), 1-10.

Thanks a lot. We added the references to the introduction of the revised version.

Reviewer #4:

The authors need to be careful on how they use terms related to "navigation" and "spatial". For example, the authors should make it clear that they are studying animal orientation, rather than animal navigation, as navigation implies an individual has the capacity to know its current place relative to a specific destination (e.g., map sense). Orientation and navigation are related, yet distinct concepts. Moreover, if the authors are studying the coding of a "goal direction", then this is consistent with orientation not navigation, since a "goal direction" can also be simply called a "preferred direction" or "preferred orientation heading", and is not a specific destination. The so-called "spatial goal" is not an actual, defined place (an actual goal, such as the overwintering sites of migratory monarch butterflies), as directional behavior to this "goal" and the "goal" itself, can be manipulated as seen in this study. As such, it is more appropriate for the authors to describe their work as further examining how animals orient, as they are examining how animals encode a preferred goal heading and how that heading is related to the animal's actual current heading during movement.

We thank the reviewer for this comment. Our aim was clearly not to present our data in the context of a specific goal (which was misleading in the previous version) but rather how a goal direction is processed. We therefore went through our manuscript and changed the wording/meaning where applicable (e.g., first sentence of our abstract and introduction).

In the neuroscience community, both terms orientation and navigation are often used in an interchangeable way. However, the reviewer is correct that navigation is used to categorize a more elaborated orientation behavior in behavioral ecology. We are happy to follow these definitions and carefully went through our manuscript and replaced the term 'navigation' by 'spatial orientation' to avoid any misinterpretation of the behavior that we studied here.

A major goal of the authors in this manuscript is to relate their results to how migratory monarch butterflies might encode goal directionality (see Introduction and Discussion). Unfortunately, as stated by the authors themselves in the Results (lines 76-77), they are not studying migratory animals at all in their work, and therefore it remains unclear how these results actually relate to the orientation mechanisms of migratory butterflies and migratory animals in general. A large body of work has clearly shown that migratory and non-migratory monarch butterflies are fundamentally different from each other, such as at the morphological, behavioral, physiological, and genetic levels. Moreover, recent work (e.g., Tenger-Trolander et al. 2019) has shown that monarchs from commercial stocks and then reared under controlled conditions (as done by the researchers in this current study; lines 230-234) do not orient in the same way when tested in flight experiments, as compared to migratory butterflies caught in the wild. Therefore, these results appear to be preliminary and have limited applicability to migratory butterflies, especially within the context of orientation during migration. The authors need to study actual migratory monarch butterflies in their experiments, whose migratory directionality is confirmed when caught in the field, in order to see if these neural mechanisms are applicable to these animals during behavior.

The reviewer raises an important point that we did not fully address in our first manuscript version. While the reviewer is correct in stating that migratory and non-migratory butterflies are behaviorally, genetically, and physiologically different, all monarch butterfly brains (even all insect brains) are equipped with an evolutionarily highly conserved central-complex circuit, which is astonishingly similar even down to single cells (e.g., Pfeiffer and Homberg 2014, *Ann Rev Entomol*) and that controls their movements in space. Thus, the same neurons can be found in migratory and non-migratory butterflies and differences in their spatial coding are limited to the neuron's tuning bandwidth while the distribution of pfd's are the same (Nguyen et al. 2021, *Proc R Soc B*). While the central complex controls menotactic orientation in non-migratory butterflies (Beetz et al. 2022, *Curr Biol*) it highly likely controls the migration in migratory butterflies (Heinze and Reppert, 2011, *Neuron*). Slight synaptic modifications of the same neurons might modify the central-complex network from a system that encodes menotactic orientation to a network that processes migration (Heinze 2017, *Curr Opin Insect Sci*; Honkanen et al. 2019, *JEB*). To control these orientation behaviors, the central complex needs to house goal-direction neurons, but their existence has so far only been predicted theoretically (e.g. Stone et al. 2017, *Curr Biol*).

Our study provides the first evidence of how the insect brain encodes a goal direction, irrespective of a specific orientation behavior. As evident from the positive comments of the other reviewers, our study is a breakthrough in our understanding of the neural principles of spatial orientation in insects and is an important step towards understanding how specific orientation behaviors, such as migration, are encoded in the insect brain. We therefore disagree with the reviewer's statement that these data are "preliminary" and would like to emphasize that this is an important paper, highly relevant to the neuroscience community.

A long-term goal of our research is indeed to understand the migratory behavior. However, if the experiments presented here are even feasible in the context of migration is unclear. Can we easily

deviate the migratory butterflies from their migratory flight direction given that their goal direction is genetically fixed? This is an important requirement to dissociate goal direction from head direction coding as demonstrated here. Do migratory butterflies even maintain a heading relative to the virtual sun indoors that we can relate to their southerly migratory direction? We suspect that we need to perform these challenging experiments in a flight simulator outdoors to study the neural mechanisms of migration. However, performing tetrode recordings outside has never been done so far and is not trivial to accomplish.

In general, as outlined above, our results on the neural principles of goal-direction coding in non-migratory butterflies combined with the highly conserved nature of the insect central complex allow us to make predictions about how migration is likely encoded in the migratory butterfly morph.

In this context, we agree with the reviewer that the role of the central complex in spatial orientation and its potential role in migration was not clear in our first draft. We therefore explicitly discuss these points in the revised draft, further emphasizing that our recordings were obtained from non-migratory butterflies. We added the following sections in the discussion to place our results into the appropriate conceptual frame.

On lines 279-284, we write: ‘Whether distance coding has any behavioral relevance in non-migratory or migratory monarch butterflies remains unclear. Like migratory birds, migratory monarch butterflies rely most likely on a stop signal rather than distance information to localize their migratory goal’⁶⁶. Thus, it would not be surprising if the here described GD neurons in monarch butterflies do not encode distance information and purely encode directional information for short-distance dispersal (non-migratory) or long-distance migration (migratory).’

*On lines 297-321, we added: ‘Due to the highly conserved nature of the central complex^{1,35,36,67} our results give deep insights into the general coding of goal-directed orientation in insects. Our recordings were obtained from non-migratory monarch butterflies that are closely related to the population of migratory monarch butterflies but lost their ability to migrate^{68,69}. Thus, in contrast to the single (southward) goal-direction set by the population of migratory monarch butterflies, the non-migratory butterflies maintain any possible goal direction with respect to a virtual sun (menotactic orientation)^{47,52}. Non-migratory monarch butterflies were ideal for our experiments because we could easily manipulate their goal directions. Despite the molecular and behavioral differences found between migratory and non-migratory monarch butterfly populations^{28,68,70-74}, the anatomy of the central-complex network in the monarch brain can be expected to be highly similar, even down to single sun compass neurons^{33,75}. Thus, differences in the orientation behaviors between the migratory and non-migratory monarch butterflies likely results from synaptic modifications of the same neurons which modify the intended goal directions^{25,36}. Therefore, our recordings allow us to predict how the tuning of the same central-complex neurons might be modified to encode long-distance migration in migratory monarch butterflies: Like in all other insects, the monarch butterfly fan-shaped body is compartmentalized into 16 columns⁷⁵. We predict that a population of GD neurons, homologous to the ones described in this study, represents the migratory southward direction within the columns of the fan-shaped body^{1,2,58}, similar to how the HD network represents a compass of heading directions across the columns of the ellipsoid body of the central complex^{3,4,56}. By changing the compass polarity through sun displacements, we might have induced a translocation of the HD representation in the butterflies’ ellipsoid body, as it has also been demonstrated in *Drosophila* 3. Contrastingly, the GD representation was unaffected by compass perturbations⁵⁸. Resetting the goal direction through aversive conditioning, however, might have induced a translocation of the GD representation across the columns of the fan-shaped body. We propose that a similar translocation of the GD representation might transform the butterfly’s southward direction into a northward one in migratory monarch butterflies before departing for their remigration in spring⁷⁶.’*

Do the animals actually change their heading in relation to the displacement (e.g., 180° sun displacement; Figure 1D) or is this view simply an artifact of the experimenter's concept of space? It would appear that the animals are not changing their heading at all (especially within the confines of the uniform experimental arena), but are actually maintaining their heading relative to their perception of the virtual sun. For example, in Figure 1D, the butterflies are keeping the position of the virtual sun to the left (left of the mean heading as denoted by the purple line). After the experimenter moved the position of the virtual sun, the butterflies then again simply kept the sun in the same position relative to them, i.e., the virtual sun to the left. Effectively, the butterflies did not change their heading relative to the virtual sun, but maintained their subjective heading. Rather than a preferred heading, which suggests a specific direction such as southwards for a migratory monarch butterfly, could this behavior, and its shift in response to moving the virtual sun, reflect a form of positive phototaxis in which monarchs fly towards light regardless of where the light source is, but with some slight skew? It seems that results from Figure 2 support this, as the aversive conditioning (electrical shocks to the butterflies) makes the butterflies fly almost directly away from the virtual sun, who then are displaying negative phototaxis in trials.

From the experimenter's view the butterflies were changing their absolute heading direction by 180° after a 180° sun displacement (allocentric frame of reference). From the butterfly's perspective, it has not changed its heading direction relative to the sun as before the displacement (i.e., it aims for the same goal direction) (egocentric frame of reference). In a recent study we have established that the monarch butterfly central complex relies on multimodal (allothetic and idiothetic) information to establish an allocentric compass (Beetz et al. 2022, *Curr Biol*), which is also true for other insects (Seelig and Jayaraman 2015 *Nature*). This means that compass neurons encode spatial orientation in an allocentric frame of reference and therefore sun displacements induce a change in the butterfly's compass polarity. This has already been shown in compass neurons of *Drosophila* whose angular tuning was affected after cue displacements (Green et al. 2019 *Nat Neurosci*). In contrast to the allocentric compass, the goal direction is represented in an egocentric frame of reference, i.e., the goal direction is encoded relative to an orientation cue (in our case the virtual sun). This is well in line with the vertebrate literature (e.g., Finkelstein et al. (2015) *Nature*; Sarel et al. (2017) *Science*), further showing how similar the neural coding of heading information is between invertebrates and vertebrates.

A phototaxis can be excluded for multiple reasons. As indicated by Fig. S1, and Figure S9, the butterflies demonstrate individual specific and arbitrary flight directions relative to the sun. This behavior defined as menotactic orientation is encoded in the central complex. In contrast, phototaxis is not controlled by the central complex (Giraldo et al. 2018, *Curr Biol*). In addition, behavioral experiments demonstrated that monarch butterflies can switch between menotactic and phototactic behavior within individual butterflies by replacing the virtual sun with a vertical stripe (Fig. 5A, Franzke et al. 2022, *JEB*). In response to the vertical stripe, the monarch butterflies show clear stripe fixation which can be explained with a simple phototaxis strategy (see also Figs. 3G,I in Franzke et al. 2022, *JEB*). In contrast to this, a virtual sun is always used for menotactic orientation (Fig. 3F, Franzke et al. 2022, *JEB*). However, there is often a trend that insects set a direction in the hemisphere of the virtual sun in indoor experiments (e.g., el Jundi et al. (2016) *Curr Biol*; Giraldo et al. 2018, *Curr Biol*). The reviewer is correct, that this was also true for the butterflies tested prior to aversive conditioning (Fig. S9, left heatmap and circular plot summarizing the mean headings before conditioning). As the heading directions which induced an electric shock were relatively large ($\pm 90^\circ$ to the goal direction), this resulted in a trend that the butterflies tend to fly in the opposite direction of the virtual sun (Fig. S9, right heatmap and circular plot summarizing the mean headings after conditioning).

Taken together, it is highly unlikely that our findings could be explained in the context of phototaxis because we would not have expected such neural tuning shifts in the central complex as this brain region is not involved in phototaxis. We thank the reviewer for pointing this out. It is indeed an important point that we did not mention. We therefore added on lines 175-181: *'After conditioning, most butterflies (13 of 17) headed into the hemisphere opposite to the virtual sun (Fig. S9). This is not surprising considering that most of the 17 butterflies set their goal direction toward the virtual sun hemisphere prior to conditioning, a trend often observed in indoor experiments^{39,52,53}. However, the butterflies' headings showed a larger distribution (Fig. S9B), than what would be expected from a negative phototactic behavior in monarch butterflies⁴⁷, suggesting that their orientation strategy remained a menotactic behavior'*

REVIEWER COMMENTS

Reviewer #1 (Remarks to the Author):

The authors have addressed my concerns and I support publication in Nature Communications. This is an elegant study that provides insight into the organization of heading, goal direction, and steering neurons in the insect brain and will provide a basis future investigations on the organization and evolution of navigation circuits across species.

Reviewer #2 (Remarks to the Author):

The authors have addressed all my comments in full. I recommend the paper for publication.

Reviewer #3 (Remarks to the Author):

The authors have done an excellent job of addressing my comments and those of the other reviewers.

My only final request is that the authors quantify what 'no hints' means at the end of the new prose:

"(lines 459-466): 'To characterize the spatial coding in the tethered butterflies, the tested animals should have ideally explored all possible heading directions uniformly. However, when the goal is to maintain a directed course and the animal's heading is thus biased towards the goal direction, as it was the case in the current study, a uniform heading representation is experimentally impossible 8 . To minimize the effect of a biased heading representation on the angular tuning, we quantified the angular tuning based on the neurons' mean firing rates instead of accumulating single spike events. Moreover, during the experiments, the butterflies had to explore each possible heading direction to allow us to calculate the spatial tuning.

We found no hints that the behavioral bias in heading affected the neural tuning."

I think readers will puzzle over why this is examined quantitatively.

Reviewer #4 (Remarks to the Author):

The authors contend in their rebuttal letter that the brains of monarch butterflies are highly conserved and that they expect the anatomy of monarch butterfly brains to be similar (lines 305-306), with differences between migrants and non-migrants predicted to be simply due to synaptic modifications of the same neurons (lines 307-308). Unfortunately, doing research with captive-reared non-migratory monarch butterflies is a problematic proxy for studying how things might work with wild migrant monarch butterflies for the following reasons.

1. As stated by the authors in their manuscript, they only used captive-reared, non-migratory monarch butterflies for their experiments. Although the authors believe that non-migratory butterflies are "ideal" for their experiments, it has been demonstrated that using such butterflies for examining monarch butterflies, in particular work that sets to explain the biology of migratory monarch butterflies specifically, is not ideal. For example, Tenger-Trolander et al. (2019), Tenger-Trolander and Kronforst (2020), and Davis et al. (2020), all show that captive-reared monarch butterflies are significantly different from wild migratory monarch butterflies in terms of oriented flight behavior, wing morphology, and physiology. As such, any work that only examines captive-reared monarch butterflies, particularly those that are in the non-migratory state, should be used quite conservatively for extrapolating or predicting on how things work for migratory monarch butterflies.

2. The authors are still underestimating the significant differences between migratory and non-migratory monarch butterflies in this revised version of the manuscript. A significant body of literature at several levels of analysis has clearly demonstrated that these two groups of butterflies are almost fundamentally different from each other, with migratory butterflies possessing a distinct migratory syndrome that shapes their entire biology.

In addition to many key behavioral and molecular differences between these two groups of butterflies (lines 303-305 of revised manuscript), migratory and non-migratory monarch butterflies are further different from each other in the following ways: a) hormonally and in endocrine regulatory mechanisms (Herman and Tatar, 2001); b) in the onset of reproduction (diapause in migratory butterflies; Oberhauser and Solensky, 2004) and longevity (Herman and Tatar, 2001); c) metabolically (Tenger-

Trolander et al., 2023); and d) in multiple morphological traits that significantly affect migratory flight behavior (Dockx, 2007; Davis, 2009; Altizer and Davis, 2010; Li et al., 2016).

Similarly, previous work by the authors themselves showed that the brains of migratory and non-migratory monarchs are also different from each other. For example, there are volumetric differences in brain regions (neuropils) between migrants and non-migrants (Heinze et al., 2013), and the encoding of orientation information in the brain (tuning width of neurons) is also different between migrants and non-migrants (Nguyen et al., 2021).

Although the current manuscript might be relevant to the neuroscience community with how a brain might encode spatial orientation information and has received positive comments, the work is “preliminary” and largely speculative when discussing the results as extending to the actual biology of migratory monarch butterflies. The authors can make predictions on how they think the brain of a migratory monarch butterfly might work, but it appears that they are overextending the interpretation of their results. As the stated long-term goal of the authors is to understand migratory behavior, based on the documented limitations of using captive-reared monarch butterflies as study organisms and the fundamental differences between migratory and non-migratory monarch butterflies, the authors should further temper their discussion of their results in the revision. I understand that the authors want to enhance the impact and generalizability of their manuscript, but they should also more fully describe the limitations of their work in the text to readers, given what has already been published.

We thank the reviewers for their positive comments. Please, find below the reviewers' points (in black) and our responses (in green). To facilitate the review process, all major changes are copied to the response-to-reviewers letter.

Reviewer 3:

The authors have done an excellent job of addressing my comments and those of the other reviewers. My only final request is that the authors quantify what 'no hints' means at the end of the new prose: "(lines 459-466): 'To characterize the spatial coding in the tethered butterflies, the tested animals should have ideally explored all possible heading directions uniformly. However, when the goal is to maintain a directed course and the animal's heading is thus biased towards the goal direction, as it was the case in the current study, a uniform heading representation is experimentally impossible⁸. To minimize the effect of a biased heading representation on the angular tuning, we quantified the angular tuning based on the neurons' mean firing rates instead of accumulating single spike events. Moreover, during the experiments, the butterflies had to explore each possible heading direction to allow us to calculate the spatial tuning. We found no hints that the behavioral bias in heading affected the neural tuning.'"

I think readers will puzzle over why this is examined quantitatively.

To avoid any misunderstanding, we changed the wording (lines 476-482): *"Despite a bias in heading representation, changes in the mean heading did not affect the angular tuning of central-complex neurons. For example, perturbations of the butterfly's compass elicited a substantial change in the mean heading (Fig. 1D) but did not affect the angular tuning of putative GD neurons (blue neuron in Fig. 1H). Similarly, the angular tuning of HD neurons was unaffected (Fig. 2G) despite substantial changes in the butterfly's mean heading elicited by aversive conditioning (Fig. 2E). This suggests that a bias in heading representations within an individual animal did not influence the angular tuning of central-complex neurons."*

Reviewer 4:

The authors contend in their rebuttal letter that the brains of monarch butterflies are highly conserved and that they expect the anatomy of monarch butterfly brains to be similar (lines 305-306), with differences between migrants and non-migrants predicted to be simply due to synaptic modifications of the same neurons (lines 307-308). Unfortunately, doing research with captive-reared non-migratory monarch butterflies is a problematic proxy for studying how things might work with wild migrant monarch butterflies for the following reasons.

We appreciate the thorough review by the reviewer. We clearly do not want to state that the anatomy of monarch butterfly brains is similar. This statement would, in our opinion, be wrong. Of course, we expect differences between non-migratory and migratory monarch butterfly brains based on different sensory inputs, a different ecology, and different behaviors – this is also not what we are writing. What we are pointing out is that **the anatomy and function of the insect central complex** is highly conserved (see Heinze, 2017, Curr Opin Insect Sci; Honkanen et al. 2019, J Exp Biol; Pfeiffer and Homberg, 2014, Ann Rev Entomol). To avoid any confusion, we changed the wording on lines 297 and 310 to emphasize that we specifically refer here to the anatomy and function of the central complex and not to the whole brain.

1. As stated by the authors in their manuscript, they only used captive-reared, non-migratory monarch butterflies for their experiments. Although the authors believe that non-migratory butterflies are "ideal" for their experiments, it has been demonstrated that using such butterflies for examining monarch butterflies, in particular work that sets to explain the biology of migratory monarch butterflies specifically, is not ideal. For example, Tenger-Trolander et al. (2019), Tenger-Trolander and Kronforst (2020), and Davis et al. (2020), all show that captive-reared monarch butterflies are significantly different from wild migratory monarch butterflies in terms of oriented flight behavior, wing morphology, and physiology. As such, any work that only examines captive-reared monarch butterflies, particularly those that are in the non-migratory state, should be used quite conservatively for extrapolating or predicting on how things work for migratory monarch butterflies.

We agree with the reviewer: stating that captive-reared monarch butterflies are “ideal” to perform behavioral experiments is not correct given the papers stated by the reviewer. In addition, they are not “ideal” to investigate the neurobiology of migration. We therefore removed the corresponding statement and clearly state the limitation of our work on lines 302 - 308:

“Because non-migratory monarch butterflies demonstrate individual specific goal directions^{68,69}, we reasoned that their goal direction coding can experimentally be controlled and manipulated under laboratory conditions. However, as non-migratory captive-reared monarch butterflies differ behaviorally⁶⁸⁻⁷⁰, morphologically⁷¹⁻⁷³, physiologically⁷², and molecularly^{74,75} from migratory monarch butterflies it remains speculative how the here presented orientation network functions in nature and how it can encode the annual migration in migratory butterfly populations.”

2. The authors are still underestimating the significant differences between migratory and non-migratory monarch butterflies in this revised version of the manuscript. A significant body of literature at several levels of analysis has clearly demonstrated that these two groups of butterflies are almost fundamentally different from each other, with migratory butterflies possessing a distinct migratory syndrome that shapes their entire biology.

In addition to many key behavioral and molecular differences between these two groups of butterflies (lines 303-305 of revised manuscript), migratory and non-migratory monarch butterflies are further different from each other in the following ways: a) hormonally and in endocrine regulatory mechanisms (Herman and Tatar, 2001); b) in the onset of reproduction (diapause in migratory butterflies; Oberhauser and Solensky, 2004) and longevity (Herman and Tatar, 2001); c) metabolically (Tenger-Trolander et al., 2023); and d) in multiple morphological traits that significantly affect migratory flight behavior (Dockx, 2007; Davis, 2009; Altizer and Davis, 2010; Li et al., 2016).

Again, we appreciate the concerns raised by the reviewer regarding the relevance of our results with respect to migration. To avoid any over-interpretation of our results, we added the following statement to the manuscript (lines 328 – 331): *“However, as the behavior^{68,69}, endocrinology⁷⁸, reproduction²⁹, longevity⁷⁸, metabolism⁷², and morphology^{73,79-81} differs drastically between migratory and non-migratory butterflies, it is crucial to perform recordings in migratory butterflies in the future to conclusively reveal how migration is controlled by the central complex network.”*

Similarly, previous work by the authors themselves showed that the brains of migratory and non-migratory monarchs are also different from each other. For example, there are volumetric differences in brain regions (neuropils) between migrants and non-migrants (Heinze et al., 2013), and the encoding of orientation information in the brain (tuning width of neurons) is also different between migrants and non-migrants (Nguyen et al., 2021).

The reviewer is correct about these papers, but they neither contradict in any way our results, nor any statement raised in our manuscript. Differences in brain volumes in higher brain regions, such as in the central complex, are often a consequence of a higher synaptic plasticity (i.e., arise from a higher number of synaptic contacts between neurons; see for an example Schmitt et al. 2016, Dev Neurobiol). Thus, a change in volume does not necessarily correlate with a higher number of neurons. To clarify our statement, we added the following sentence to manuscript (lines 311 -314): *‘Differences in the coding of goal directions between the migratory and non-migratory monarch butterflies had been discussed to underlie synaptic modifications of the same neurons^{25,36}. Such synaptic modifications may explain volumetric differences of some brain regions in migratory and non-migratory monarch butterflies⁷⁷.’*

The physiological differences described in Nguyen et al. (2021) were found in neurons that carry sensory, visual information into the central complex (input neurons termed TL neurons) and suggest that the input into the central complex seems to differ with respect to visual sensitivity (in migratory animals the TL neurons exhibit a higher angular sensitivity). However, this higher sensitivity is not expected to have **any effect on the functionality of the neuron types** studied here (namely HD, GD, and steering neurons).

Although the current manuscript might be relevant to the neuroscience community with how a brain

might encode spatial orientation information and has received positive comments, the work is “preliminary” and largely speculative when discussing the results as extending to the actual biology of migratory monarch butterflies. The authors can make predictions on how they think the brain of a migratory monarch butterfly might work, but it appears that they are overextending the interpretation of their results. As the stated long-term goal of the authors is to understand migratory behavior, based on the documented limitations of using captive-reared monarch butterflies as study organisms and the fundamental differences between migratory and non-migratory monarch butterflies, the authors should further temper their discussion of their results in the revision. I understand that the authors want to enhance the impact and generalizability of their manuscript, but they should also more fully describe the limitations of their work in the text to readers, given what has already been published.

Taken together, as evident from our previous points, we agree with the reviewer. Our aim is not to over-interpret our results in any way, especially giving the reader the wrong impression that we investigated the migration of monarch butterflies. We also do not underestimate the substantial differences that had been found in migratory and non-migratory monarch butterflies in previous studies. Working on an animal that is considered a non-model organism in neuroscience, we highly appreciate the papers cited by the reviewer and are fully aware of how dramatic an animal’s behavior can be affected by the behavioral state/ rearing conditions. We carefully went through our paper and modified an additional sentence that previously could have been interpreted as if we underestimated the differences between migratory and non-migratory butterflies. We added all citations and comments raised by the reviewer (see points above). We thank the reviewer and believe that the manuscript in its current form emphasizes all limitations with respect to generalizability of our results.